# MMGP: a Mesh Morphing Gaussian Process-based machine learning method for regression of physical problems under non-parameterized geometrical variability

**Fabien Casenave    Brian Staber    Xavier Roynard**
Safran Tech, Digital Sciences & Technologies
78114 Magny-Les-Hameaux, France
{fabien.casenave, brian.staber, xavier.roynard}@safrangroup.com

## Abstract

When learning simulations for modeling physical phenomena in industrial designs, geometrical variabilities are of prime interest. While classical regression techniques prove effective for parameterized geometries, practical scenarios often involve the absence of shape parametrization during the inference stage, leaving us with only mesh discretizations as available data. Learning simulations from such mesh-based representations poses significant challenges, with recent advances relying heavily on deep graph neural networks to overcome the limitations of conventional machine learning approaches. Despite their promising results, graph neural networks exhibit certain drawbacks, including their dependency on extensive datasets and limitations in providing built-in predictive uncertainties or handling large meshes. In this work, we propose a machine learning method that do not rely on graph neural networks. Complex geometrical shapes and variations with fixed topology are dealt with using well-known mesh morphing onto a common support, combined with classical dimensionality reduction techniques and Gaussian processes. The proposed methodology can easily deal with large meshes without the need for explicit shape parameterization and provides crucial predictive uncertainties, which are essential for informed decision-making. In the considered numerical experiments, the proposed method is competitive with respect to existing graph neural networks, regarding training efficiency and accuracy of the predictions.

## 1   Introduction

Many problems in science and engineering require solving complex boundary value problems. Most of the time, we are interested in solving a partial differential equation (PDE) for multiple values of input parameters such as material properties, boundary conditions, initial conditions, or geometrical parameters. Traditional numerical methods such as the finite element method, finite volume method, and finite differences require fine discretization of time and space in order to be accurate. As a result, these methods are often computationally expensive, especially when the boundary value problem needs to be repeatedly solved for extensive exploration of the input parameters space. To overcome this issue, machine and deep learning have been leveraged for various tasks in computational physics, namely, solving and learning solutions to PDEs [37, 56, 65, 81], accelerating linear solvers [5, 34], reduced-order modeling [49], domain decomposition [41], closure modeling [52], and topology optimization [74], to name a few. As reported in the review papers [13, 17, 75], most of the recent advances have been relying on deep neural networks for their flexibility and expressiveness. In this work, we focus on learning simulations of physical phenomena, that are discretized on a non-

37th Conference on Neural Information Processing Systems (NeurIPS 2023).

parameterized unstructured mesh. In this situation, traditional machine learning approaches cannot easily be leveraged as the inputs of the problem are given by graphs with different numbers of nodes and edges. In contrast, deep learning models such as graph neural networks (GNNs) [67] can easily overcome this limitation thanks to their ability to operate on meshes with different resolutions and topologies. While GNNs show promising results and their flexibility is highly appealing, they still suffer from a few shortcomings that prevent their deployement in engineering fields where decisions involve high stakes. Training GNNs usually requires large datasets and computational resources, and predicting their uncertainties is still an open and challenging problem of its own [32].

We propose a novel methodology, called Mesh Morphing Gaussian Process (MMGP), that relies on standard and well-known morphing strategies, dimensionality reduction techniques and finite element interpolation for learning solutions to PDEs with non-parameterized geometric variations. In contrast to deep learning methods, such as GNNs, the model can easily and efficiently be trained on CPU hardware and predictive uncertainties are readily available. Our method shares some limitations with any machine learning regressor for PDE systems: (i) within the predictive uncertainties, our method produces predictions with an accuracy lower than the reference simulations, (ii) unlike many methods used in reference simulators, like the finite element method, our method provides no guaranteed error bounds and (iii) our method requires a well sampled training dataset, which has a certain computational cost, so that the workflow becomes profitable only for many-query contexts where the inference is called a large number of times. Regarding (i), rough estimates may be sufficient in preproject phases, and accuracy can be recovered by using the prediction as an initialization in the reference simulator, or by allowing the designer to run the reference simulator on the identified configuration if the regressor is used in an optimization task.

We start by providing the background and assumptions of our method while mentioning some related works in Section 2. Then, the proposed methodology is detailed in Section 3. Three numerical experiments are presented in Section 4. Finally, a conclusion is given in Section 5.

## 2 Preliminaries and related works

**Notations.** Vectors and matrices are denoted with bold symbols. The entries $i$ of a vector $\mathbf{v}$ and $i, j$ of a matrix $\mathbf{M}$ are respectively denoted $v_i$ and $M_{i,j}$. The i-th row of a matrix $\mathbf{M}$ is denoted by $\mathbf{M}_i$.

**Background.** Let $\mathcal{U}_{\mathrm{true}} : \Omega \to \mathbb{R}^d$ be a solution to a boundary value problem, where $\Omega \subset \mathbb{R}^{d_\Omega}$ denotes the physical domain of the geometry under consideration, and $d_\Omega = 2$ or 3. The domain $\Omega$ is discretized into a conformal mesh $\mathcal{M}$ as $\mathcal{M} = \cup_{e=1}^{N_e} \Omega_e$. In traditional numerical approaches such as the finite element method [82], an approximation $\mathcal{U}$ of the solution $\mathcal{U}_{\mathrm{true}}$ is sought in the finite-dimensional space spanned by a family of trial functions, $\{\varphi_I(\mathbf{x})\}_{I=1}^N$, supported on the mesh $\mathcal{M}$:

$$\mathcal{U}_k(\mathbf{x}) = \sum_{I=1}^N U_{k,I}\varphi_I(\mathbf{x}), \quad k = 1, \ldots, d, \tag{1}$$

where $N$ is the total number of nodes in the mesh $\mathcal{M}$, $\mathbf{U} \in \mathbb{R}^{d \times N}$ is the discretized solution (featuring $d$ fields), and $\mathbf{x} \in \mathbb{R}^{d_\Omega}$ denotes the spatial coordinates. For simplicity of the presentation and without loss of generality, we consider the particular case of a Lagrange $\mathbb{P}_1$ finite element basis, so that the solution is uniquely determined by its value at the nodes of $\mathcal{M}$. In this setting, the basis $\{\varphi_I\}_{I=1}^N$ spans the space $\{v \in \mathcal{C}^0(\mathcal{M}) : v|_{\Omega_e} \in \mathbb{P}_1, \forall \Omega_e \in \mathcal{M}\}$, and the discretized solution $\mathbf{U}$ is determined by solving the discretized weak formulation of the underyling boundary value problem. This problem also depends on some parameters $\boldsymbol{\mu} \in \mathbb{R}^p$, such as material properties and boundary conditions. It is assumed that there are scalar output quantities of interest $\mathbf{w} \in \mathbb{R}^q$ that depend on the discretized solution $\mathbf{U}$, and possibly on $\mathcal{M}$ and $\boldsymbol{\mu}$. We restrict ourselves to stationary, time-independent, scalars and fields of interest, which still falls in the scope of many industrial problems of interest. The learning task that we consider herein consists in learning the mapping

$$\mathcal{F} : (\boldsymbol{\mu}, \mathcal{M}) \mapsto (\mathbf{U}, \mathbf{w}). \tag{2}$$

For this purpose, it is assumed that we are given a training set of size $n$ made of input pairs $(\boldsymbol{\mu}^i, \mathcal{M}^i)$ of parameters and meshes, and output pairs $(\mathbf{U}^i, \mathbf{w}^i)$ of discretized fields and scalars. Each input mesh $\mathcal{M}^i$ has a number of nodes denoted by $N^i$, and corresponds to a finite element discretization of an input geometry $\Omega^i$. The associated discretized solution $\mathbf{U}^i$ is a matrix of size $(d \times N^i)$. For

any $i = 1, \ldots, n$, the mesh $\mathcal{M}^i$ can be represented as an undirected graph $G^i = (V^i, E^i)$, where $V^i$ denotes the set of nodes and $E^i$ is the set of edges.

**Assumptions and limitations.**    We assume that the observed input geometries, $\Omega^1, \ldots, \Omega^n$, share a common topology. The parameterization that generates the input geometries is unknown, and we are left with the associated finite element meshes $\mathcal{M}^1, \ldots, \mathcal{M}^n$. Being the discretization of physical domains involved in a boundary value problem, the input meshes inherit important features such as boundary conditions applied to subsets of nodes and elements. In finite element methods, error estimates strongly depend on the quality of the mesh [66]. Hence, in our context, it is assumed that the input meshes exhibit good quality in terms of elements aspect ratios and node densities, adapted to the regularity of the fields of interest. Our focus centers on the design optimization of industrial components with respect to specific physical phenomena. Consequently, we assume precise control over the geometry, free from any noise. Additionally, the employed geometrical transformations are constrained to avoid extreme distortions, as they are selected from sets of admissible designs that adhere to limitations on mass, volume, and mechanical resistance.

**Related works.**    In recent years, there has been a substantial focus on advancing neural networks to emulate solutions to physical systems, either through the integration of domain-specific knowledge [44] or by devising efficient architectures for GNNs [63]. GNNs learn the mapping $\mathcal{F}$ by relying on the message passing framework introduced by Gilmer et al. [33] and extended by Battaglia et al. [11]. In the context of physical systems, only a few contributions address non-parameterized geometric variabilities. The early work of Baque et al. [10] explores the use of GNNs to emulate physics-based simulations in the presence of geometric variabilities by relying on geodesic convolutions [58, 61]. More recently, Pfaff et al. [63] develop the MeshGraphNets (MGNs) model, a GNN that updates nodes and edges features in order to learn time-dependent simulations. Most notably, the model can handle various physics, three-dimensional problems, and non-parameterized geometric variabilities. Fortunato et al. [28] introduce MultiScale MGNs that relies on two different mesh resolutions in order to overcome the computational cost of the message passing algorithm on dense meshes, and to increase the accuracy of MGNs. The efficiency of MGNs has been illustrated by Allen et al. [7] for inverse problems, and by Harsch et al. [39] for time-independent systems. There exist several variants of such GNNs for learning mesh-based solutions. A multi-scale GNN that learns from multiple mesh resolutions has been proposed in Lino et al. [53] and is illustrated on two dimensional PDEs with geometric variabilities. Lino et al. [54] also devise a multi-scale and rotation-equivariant GNN that extrapolates the time evolution of the fluid flow, and Cao et al. [19] propose a novel pooling strategy that prevents loss of connectivity and wrong connections in multi-level GNNs. Regarding morphing strategies, Gao et al. [30] and Li et al. [51] deform irregular meshes into a reference one in order to learn solution of PDEs, but rely on complex coordinate transformation to compute a physical residual-based loss in the reference domain, and on input meshes with equal number of nodes. It is worth emphasizing that while the aforementioned works show promising results, they do not provide predictive uncertainties. There exist several methods for quantifying the uncertainties of deep neural networks [32], but it remains an open problem to provide well calibrated uncertainty estimates at a reasonable computational cost.

## 3   MMGP methodology

The proposed methodology is based on two main ingredients that allow us to leverage classical machine learning methods for regression tasks in the context of non-parameterized geometrical variability: (i) the data is pretreated by morphing each input mesh into a reference shape, and resorting to finite element interpolation to express all fields of interest on a common mesh of this reference shape, and (ii) a low-dimensional embedding of the geometries is built by considering the coordinates of the nodes as a continuous input field over the meshes. Formally, the proposed methodology consists in constructing a graph kernel by relying on three well chosen transformations such that the transformed inputs can be compared with any classical kernel functions defined over Euclidean spaces. Figure 1 illustrates the proposed strategy for a two-dimensional problem where we aim at predicting output fields of interest. The first transformation morphs the input mesh onto a chosen common shape. The second transformation performs a finite element interpolation on the chosen reference mesh of the common shape. Finally, a dimensionality reduction technique is applied to obtain low-dimensional embeddings of the inputs and outputs. These three steps are

all deterministic and described in the following subsections. The proposed kernel function can be plugged into any kernel method. Herein, we rely on Gaussian process regression in order to learn steady-state mesh-based simulations in computational fluid and solid mechanics.

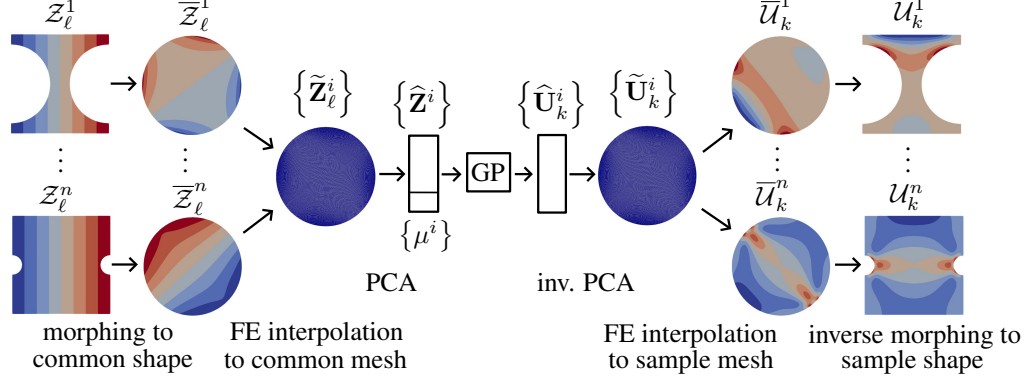

Figure 1: Illustration of the MMGP inference workflow for the prediction of an output field of interest. The lower rectangle in the illustration of the input of the GP represents the scalar inputs $\mu^i$.

## 3.1 Deterministic preprocessings of the input meshes and fields of interest

In this section, we describe the methodology for building low-dimensional representations of the input meshes and output fields.

**Mesh morphing into a reference shape $\overline{\Omega}$.** Each input mesh $\mathcal{M}^i$, $i = 1, \dots, n$, is morphed onto a mesh $\overline{\mathcal{M}}^i$ associated to a fixed reference shape $\overline{\Omega}$. The morphed mesh has the same number of nodes and same set of edges as the initial mesh $\mathcal{M}^i$, but their spatial coordinates differ. In this work, we consider two morphing algorithms, namely, Tutte's barycentric mapping [71] onto the unit disk, and the Radial Basis Function (RBF) morphing [9, 21] onto a chosen reference shape. Regardless of the morphing algorithm, physical features inherited from the boundary value problem are carefully taken into account. More precisely, points, lines and surfaces of importance in the definition of the physical problem are mapped onto their representant on the reference shape. Doing so, rigid body transformations that may occur in the database are corrected in the mesh morphing stage, and boundary conditions of same nature are matched together.

**Common mesh $\overline{\mathcal{M}}_c$ of the reference shape $\overline{\Omega}$.** At this stage, it should be noted that although the morphed meshes $\overline{\mathcal{M}}^i$ are associated to a common reference shape $\overline{\Omega}$, they do not share the same nodes and edges. This prevents us from measuring similarities between the input meshes and output fields with classical techniques. A strong advantage of the finite element method is that it provides accurate solution fields with a continuous description over the mesh, and a natural way to transfer fields from one mesh to another. This motivates us to introduce a common mesh $\mathcal{M}_c$ of the reference shape $\overline{\Omega}$, as a common support for all the sample fields data. A possibility is to choose an input mesh in the training set, e.g. $\mathcal{M}^1$, and define $\overline{\mathcal{M}}_c$ as its morphing onto the chosen reference shape. The aim is twofold. First, it allows us to express the output fields on the common morphed mesh $\overline{\mathcal{M}}_c$, leading to vector representations of same sizes. Second, the coordinates fields of the meshes $\overline{\mathcal{M}}^i$ are also transferred onto this common mesh in order to build a shape embedding. These procedures rely on classical finite element interpolation that is described in the rest of this section.

**Transporting the fields of interest on the common mesh $\overline{\mathcal{M}}_c$.** The discretized solution $\mathbf{U}^i$, $i = 1, \dots, n$, is first transferred on the morphed mesh $\overline{\mathcal{M}}^i$ as follows:

$$\overline{\mathcal{U}}_k^i(\overline{\mathbf{x}}) = \sum_{I=1}^{N^i} U_{k,I}^i \overline{\varphi}_I^i(\overline{\mathbf{x}}) \, , \, , \quad k = 1, \dots, d \, , \tag{3}$$

where $\{\overline{\varphi}_I^i\}_{I=1}^{N^i}$ is the finite element basis associated to the morphed mesh $\overline{\mathcal{M}}^i$. The transported fields $\overline{\mathcal{U}}_k^1, \dots, \overline{\mathcal{U}}_k^n$ share the same geometric support (the reference shape). This implies that they can be

interpolated onto the common mesh $\overline{\mathcal{M}}_c$ using the finite element interpolation operator $P$ defined as:

$$P(\overline{\mathcal{U}}_k^i)(\overline{\mathbf{x}}) = \sum_{J=1}^{N_c} \overline{\mathcal{U}}_k^i(\overline{\mathbf{x}}_J^c)\overline{\varphi}_J^c(\overline{\mathbf{x}}) = \sum_{I=1}^{N^i}\sum_{J=1}^{N_c} U_{k,I}^i \overline{\varphi}_I^i(\overline{\mathbf{x}}_J^c)\overline{\varphi}_J^c(\overline{\mathbf{x}}),\tag{4}$$

where $\{\overline{\varphi}_I^c\}_{I=1}^{N^i}$ is the finite element basis associated to $\overline{\mathcal{M}}_c$, $\overline{\mathbf{x}}_J^c$ is the coordinates of the $J$-th node of $\overline{\mathcal{M}}_c$ and $\overline{\mathcal{U}}_k^i(\overline{\mathbf{x}}_J^c)$ is evaluated using Equation (3). We are now in the much more favorable situation where all the fields of interest are expressed on a common mesh $\overline{\mathcal{M}}_c$. More specifically, for each field of interest $k$ and input mesh $\mathcal{M}^i$, we let $\widetilde{\mathbf{U}}_k^i \in \mathbb{R}^{N_c}$ be the transported output fields onto the common mesh, such that $\widetilde{\mathbf{U}}_{k,I}^i = \overline{\mathcal{U}}_k^i(\overline{\mathbf{x}}_I)$. In this setting, the vector representations of the output fields now have the same sizes $N_c$. Notice that the derivation of the finite element interpolation is identical with higher-order Lagrange finite elements.

**Transporting the coordinates fields on the common mesh $\overline{\mathcal{M}}_c$.** The same procedure can be applied to the coordinate fields of the input meshes in order to build a shape embedding of the input meshes. Let $\mathcal{Z}_\ell^i$ be the $\ell$-th component of the coordinate field over the mesh $\mathcal{M}^i$, $\ell = 1,\ldots,d_\Omega$. Using the finite element basis associated to the mesh $\mathcal{M}^i$, the coordinates fields can be written as

$$\mathcal{Z}_\ell^i(\mathbf{x}) = \sum_{I=1}^{N^i} Z_{\ell,I}^i \varphi_I^i(\mathbf{x}),$$

where $Z_{\ell,I}^i$ denotes the $\ell$-th of the coordinates of the node $I$ in the mesh $\mathcal{M}^i$. Notice that the notation $Z_{\ell,I}^i$ is preferred to $x_{\ell,I}^i$, since it denotes here the degrees of freedom of the coordinate fields defined over $\mathcal{M}^i$, whose continuity property is essential for the finite element interpolation stage. Then, in the same fashion as for the fields of interest, the coordinates fields are transferred on the morphed mesh $\overline{\mathcal{M}}_i$ and interpolated on the common mesh $\overline{\mathcal{M}}_c$ using the operator $P$ given by Equation (4). For each coordinate $\ell = 1,\ldots,d_\Omega$ and input mesh $\mathcal{M}^i$, we have the common representations $\widetilde{\mathbf{Z}}_\ell^i \in \mathbb{R}^{N_c}$ of the coordinate fields on the common mesh $\overline{\mathcal{M}}_c$.

**Dimensionality reduction.** At this stage, the input coordinates fields of the meshes and the output fields are expressed on the same common mesh $\overline{\mathcal{M}}_c$, and can be compared using standard machine learning techniques. We propose to build low-dimensional embeddings of these quantities using Principal Component Analysis (PCA). For each output field, PCA is applied to the set of observations $\{\widetilde{\mathbf{U}}_k^i\}_{i=1}^n$, leading to the fields low-dimensional embeddings that we denote by $\{\widehat{\mathbf{U}}_k^i\}_{i=1}^n$. Similarly, PCA is applied to concatenated transported coordinate fields, $\{(\widetilde{\mathbf{Z}}_1^i,\ldots,\widetilde{\mathbf{Z}}_{d_\Omega}^i)\}_{i=1}^n$, leading to low-dimensional embeddings of the input geometries $\{\widehat{\mathbf{Z}}^i\}_{i=1}^n$, that we refer to as the shape embeddings.

## 3.2 MMGP training

Once the operations of mesh morphing, finite element interpolation on a common mesh and dimensional reduction described in the previous subsection have been carried out, we are left with reduced-size objects of same dimension. Let $\{\mathbf{X}^i\}_{i=1}^n \in \mathbb{R}^{l_{\mathbf{z}}+p}$, where $p$ the number of nongeometrical parameters and $l_{\mathbf{Z}}$ is the size of shape embedding, be such that $\mathbf{X}^i = (\widehat{\mathbf{Z}}^i, \boldsymbol{\mu}^i)$. Denoting $l_{\mathbf{U}_k}$ the size of the embedding of field $\mathbf{U}_k$, the machine learning task given by Equation (2) can be approximated by the following set of scalar and vector regression problems:

$$\widehat{\mathcal{F}}_{\text{scalar},m}: \qquad \mathbf{X}^i \mapsto w_m^i \in \mathbb{R}, \qquad\qquad m = 1,\ldots,q,\tag{5a}$$

$$\widehat{\mathcal{F}}_{\text{vector},k}: \qquad \mathbf{X}^i \mapsto \widehat{\mathbf{U}}_k^i \in \mathbb{R}^{l_{\mathbf{U}_k}}, \qquad\qquad k = 1,\ldots,d.\tag{5b}$$

Gaussian processes can be trained in a classical fashion to address the regression problems (5a)-(5b).

**MMGP for a scalar output.** Let $\mathcal{D} = \{(\mathbf{X}^i, w_{m_0}^i)\}_{i=1}^n$ be a training dataset for one of the problems given by Equation (5a), *i.e.* for the $m_0$-th output scalar. It can be shown by standard conditioning [76] that the posterior mean and variance of the prediction on some given test input $\mathbf{X}^\star$ are given by

$$\mathbb{E}[w^\star] = \mathbf{k}_\star^T(\mathbf{K}+\sigma^2\mathbf{I})^{-1}\mathbf{w}_{m_0},$$
$$\mathbb{V}[w^\star] = K_{\star\star} - \mathbf{k}_\star^T(\mathbf{K}+\sigma^2\mathbf{I})^{-1}\mathbf{k}_\star,$$

where $\mathbf{w}_{m_0} = \{w_{m_0}^i\}_{i=1}^n$, and $\mathbf{K}$ is the Gram matrix such that $K_{i,j} = c(\mathbf{X}^i, \mathbf{X}^j)$ for $1 \le i, j \le n$, the vector $\mathbf{k}_\star$ such that $k_{\star j} = c(\mathbf{X}^\star, \mathbf{X}^j)$, and the scalar $K_{\star\star} = c(\mathbf{X}^\star, \mathbf{X}^\star)$, with $c$ denoting the chosen kernel function which lengthscales are optimized, and $\sigma$ denotes the optimized nugget parameter. This training procedure is repeated for the $q$ scalar outputs.

**MMGP for an output field.** Let $\mathcal{D} = \{(\mathbf{X}^i, \widehat{\mathbf{U}}_{k_0}^i)\}_{i=1}^n$ be a training dataset for one of the problems given by Equation (5b), *i.e.* for output field $k_0$. A multioutput GP is first trained to predict the output embeddings $\widehat{\mathbf{U}}_{k_0}$. The predictions of the GP are then decoded with the inverse PCA mapping, and morphed back to the original input mesh $\mathcal{M}^i$. Due to this last nonlinear operation, the posterior distribution of the output field of interest $\mathbf{U}_{k_0}$ is no longer Gaussian. The predictive uncertainties are thus obtained through Monte Carlo simulations. This training procedure is repeated for each of the $d$ output fields.

### 3.3 Properties of the methodology

The sequence of preprocessing operations, including mesh morphing, finite element interpolation, and PCA, leads to a non-linear dimensionality reduction. Leveraging these deterministic processes reduces the burden on the machine learning stage, potentially necessitating fewer training examples to achieve robust model performance on complex mesh-based data. In the numerical experiments presented in Section 4, the morphing technique is chosen a priori, with ongoing research focused on optimizing this morphing to minimize the number of PCA modes, which leads to a highly nonlinear dimensionality reduction stage that is finely tuned to the specific characteristics of the data.

Gaussian process regression between the input and output embeddings has several advantages. From a theoretical perspective, there exists conditions on the features of a continuous kernel so that it may approximate an arbitrary continuous target function [59]. Gaussian processes also come with built-in predictive uncertainties, that are marginally valid under the a priori Gaussian assumption. Nevertheless, the proposed methodology can be combined with any other regressor such as a deep neural network instead of the Gaussian process.

For clarity of the presentation, the MMGP methodology is illustrated with very simple, if not the simplest, morphing and dimensionality reduction techniques. Alternatives are possible for each algorithm brick. In particular, the fixed topology restriction may be lifted with other morphing algorithms, see Appendix B for more details.

## 4 Numerical experiments

Three regression problems are considered in order to assess the efficiency of the proposed methodology. The chosen datasets are first described in Section 4.1. The experimental setup is summarized in Section 4.2, and the results are discussed in Section 4.3.

### 4.1 Datasets

Three datasets in computational fluid and solid mechanics are considered, described below, and summarized in Table 1. All the considered datasets involve geometric variabilities and meshes with possibly different number of nodes and edges. Additional details can also be found in Appendix A.

Table 1: Summary of the considered datasets with $d_\Omega$: dimension of the physical problem, $p$: number of input scalars, $d$: number of output fields, $m$: number of output scalars.

| Datasets | train/test sizes | $d_\Omega$ | $p$ | $d$ | $m$ | Avg. # nodes |
|---|---|---|---|---|---|---|
| Rotor37 | $1000/200$ | 3 | 2 | 2 | 4 | $29,773$ |
| Tensile2d | $500/200$ | 2 | 6 | 6 | 4 | $9,425$ |
| AirfRANS | $800/200$ | 2 | 2 | 3 | 2 | $179,779$ |
| AirfRANS-remeshed | $800/200$ | 2 | 2 | 3 | 2 | $19,527$ |

`Rotor37` **dataset.** We consider a 3D compressible steady-state Reynold-Averaged Navier-Stokes (RANS) simulation solved with elsA [18] using the finite volumes method. The inputs are given

by a mesh representing the surface of a 3D compressor blade [8], and two additional parameters that correspond to an input pressure and a rotation speed. The outputs of the problem are given by 4 scalars (massflow $m$, compression rate $\tau$, isentropic efficiency $\eta$, polyentropic efficiency $\gamma$), and 2 fields on the surface of the input mesh (temperature $T$, pressure $P$).

`Tensile2d` **dataset.** The second dataset corresponds to a 2D quasi-static problem in solid mechanics. The geometrical support consists of a 2D square, with two half-circles that have been cut off in a symmetrical manner. The inputs are given by a mesh, a pressure applied on the upper boundary, and 5 material parameters modeling the nonlinear elastoviscoplastic law of the material [50]. The boundary value problem is solved with the finite element method and the `Z-set` software [60]. The outputs of the problem are given by 4 scalars ($p_{\max}$, $v_{\max}$, $\sigma_{22}^{\max}$, and $\sigma_v^{\max}$) and 6 fields of interest ($u$, $v$, $p$, $\sigma_{11}$, $\sigma_{12}$, and $\sigma_{22}$). For the sake of brevity, the reader is referred to Appendix A for a description of these quantities.

`AirfRANS` **dataset.** The last dataset is made of 2D incompressible RANS around NACA profiles and taken from Bonnet et al. [14]. The inputs are given by a mesh of a NACA profile and two parameters that correspond to the inlet velocity and the angle of attack. The outputs are given by 2 scalars (drag $C_D$ and lift $C_L$ coefficients), and 3 fields (the two components of the fluid velocity $u$ and $v$, and the pressure field $p$). An additional version of this dataset is also considered, where the input meshes have been coarsened using the MMG remesher [1], and the output fields have been transferred to the coarsened meshes. The output scalars are unchanged. Illustrations of the input meshes can be found in the original paper [14].

### 4.2 Experimental setup

**Morphings.** The Tutte's barycentric mapping onto the unit disk is used for morphing the meshes in the `Rotor37` and `Tensile2d` datasets. The input meshes in the `AirfRANS` dataset are morphed onto the first mesh using RBF.

**PCA embeddings.** Embeddings of sizes 32 and 64 are retained for respectively the spatial coordinates and output fields in the `Rotor37` and `AirfRANS` datasets. Smaller embeddings of sizes 8 are considered for both the spatial coordinates and output fields in the `Tensile2d` dataset. Note that for the `Tensile2d` and `AirfRANS`, a more effective variant of PCA has been used, which can easily deal with very large meshes (see, *e.g.*, [24, 36]), with up to hundreds of millions of degrees of freedom. Details about this variant can be found in Appendix C.

**Gaussian processes.** Anisotropic Matern-5/2 kernels and zero mean functions are used for all the Gaussian processes priors. The lengthscales and nugget parameter are optimized by maximizing the marginal log-likelihood function with a L-BFGS algorithm and 10 random restarts using the `GPy` package [35].

**Baselines.** The performance of MMGP is compared with two baselines, namely, a graph convolutional neural network (GCNN) with a UNet-type architecture [29] and the GeneralConv [79], and MeshGraphNets (MGN) [63]. The hyperparameters of the GNNs are chosen by relying on a grid search. The GCNN and MGN models are implemented with `PyTorch Geometric` [25] and `DGL` [73], respectively. Additional details about the architectures and hyperparameters can be found in Appendix D. Due to the sizes of the input meshes in the `AirfRANS` dataset, the considered GNN-based baselines are prohibitively expensive. Similarly to the work of [28], the GNNs are trained using coarsened input meshes as described in Section 4.1. The output fields predicted on the coarse meshes are then transferred back on the original fine meshes thanks to finite element interpolation.

**Evaluation metrics.** Accuracy of the trained models is assessed by computing relative RMSE errors. Let $\{\mathbf{U}_{\mathrm{ref}}^i\}_{i=1}^{n_\star}$ and $\{\mathbf{U}_{\mathrm{pred}}^i\}_{i=1}^{n_\star}$ be respectively test observations and predictions of a given field of interest. The relative RMSE considered herein is defined as

$$\mathrm{RRMSE}_f(\mathbf{U}_{\mathrm{ref}}, \mathbf{U}_{\mathrm{pred}}) = \left( \frac{1}{n_\star} \sum_{i=1}^{n_\star} \frac{\frac{1}{N^i}\|\mathbf{U}_{\mathrm{ref}}^i - \mathbf{U}_{\mathrm{pred}}^i\|_2^2}{\|\mathbf{U}_{\mathrm{ref}}^i\|_\infty^2} \right)^{1/2},$$

where it is recalled that $N^i$ is the number of nodes in the mesh $\mathcal{M}^i$, and $\max(\mathbf{U}_{\text{ref}}^i)$ is the maximum entry in the vector $\mathbf{U}_{\text{ref}}^i$. Similarly for scalar outputs, the following relative RMSE is computed:

$$\text{RRMSE}_s(\mathbf{w}_{\text{ref}}, \mathbf{w}_{\text{pred}}) = \left( \frac{1}{n_\star} \sum_{i=1}^{n_\star} \frac{|w_{\text{ref}}^i - w_{\text{pred}}^i|^2}{|w_{\text{ref}}^i|^2} \right)^{1/2}.$$

Given that the input meshes may have different number of nodes, the coefficients of determination $Q^2$ between the target and predicted output fields are computed by concatenating all the fields together. For each of the considered regression problems, training is repeated 10 times in order to provide uncertainties over the relative RMSE and $Q^2$ scalar regression coefficients.

### 4.3 Results and discussion

**Predictive performance.** The relative RMSE and $Q^2$ scalar regression coefficients are reported in Table 2 for all the considered experiments. While the GNN-based baselines achieve good performance, the MMGP model consistently outperforms them with lower errors. It is worth emphasizing that the field $p$ and scalar $p_{\max}$ of the `Tensile2d` dataset are particularly challenging, see Appendix E.2 for more details. They represent the accumulated plasticity in the mechanical piece, which are non-zero for a small fraction of the training set. Figure 2 shows the graphs of the output scalars predictions versus the targets on test set for the `Tensile2d` case. Figure 3 also shows examples of fields prediction in the case of the `AirfRANS` problem. The MMGP model is able to accurately reproduce the output fields, with relative errors mostly located near the tips of the airfoils.

Table 2: Means and standard deviations (gray) of the relative RMSE and $Q^2$ scalar regression coefficients for all the considered datasets and quantities of interest (QoI) (best is **bold**).

| | RRMSE | | | $Q^2$ | | |
|---|---|---|---|---|---|---|
| QoI | GCNN | MGN | MMGP | GCNN | MGN | MMGP |
| Rotor37 dataset | | | | | | |
| $m$ | 4.4e-3 (5e-4) | 5.4e-3 (7e-5) | **5.0e-4** (3e-6) | 0.9816 (4e-3) | 0.9720 (5e-4) | **0.9998** (3e-6) |
| $p$ | 4.4e-3 (5e-4) | 5.3e-3 (7e-5) | **4.8e-4** (1e-6) | 0.9803 (5e-3) | 0.9710 (9e-4) | **0.9998** (2e-6) |
| $\eta$ | 3.1e-3 (7e-4) | 7.2e-3 (7e-5) | **5.0e-4** (3e-6) | 0.9145 (4e-2) | 0.5551 (2e-3) | **0.9979** (1e-6) |
| $\gamma$ | 2.9e-3 (6e-4) | 6.5e-3 (2e-5) | **4.6e-4** (2e-7) | 0.9068 (4e-2) | 0.5257 (2e-3) | **0.9977** (2e-6) |
| $P$ | 1.7e-2 (8e-4) | 1.7e-2 (2e-3) | **7.2e-3** (5e-4) | 0.9863 (1e-3) | 0.9866 (3e-3) | **0.9973** (4e-4) |
| $T$ | 3.9e-3 (1e-4) | 1.4e-2 (2e-3) | **8.2e-4** (1e-5) | 0.9930 (5e-4) | 0.9956 (1e-3) | **0.9997** (1e-5) |
| Tensile2d dataset | | | | | | |
| $p_{\max}$ | 1.6e-0 (7e-1) | **2.7e-1** (4e-2) | 6.6e-1 (3e-1) | 0.4310 (2e-1) | 0.6400 (2e-1) | **0.9435** (2e-2) |
| $v_{\max}$ | 4.4e-2 (7e-3) | 5.8e-2 (2e-2) | **5.0e-3** (3e-5) | 0.9245 (3e-2) | 0.9830 (1e-2) | **0.9999** (2e-5) |
| $\sigma_{22}^{\max}$ | 3.1e-3 (7e-4) | 4.5e-3 (1e-3) | **1.7e-3** (2e-5) | 0.9975 (1e-3) | 0.9958 (1e-3) | **0.9993** (2e-5) |
| $\sigma_v^{\max}$ | 1.2e-1 (4e-2) | 2.4e-2 (9e-3) | **5.0e-3** (3e-5) | 0.9723 (2e-2) | 0.9801 (1e-2) | **0.9997** (7e-6) |
| $u$ | 4.5e-2 (1e-2) | 1.5e-2 (1e-3) | **3.4e-3** (4e-5) | 0.9623 (2e-2) | 0.9270 (1e-2) | **0.9997** (6e-6) |
| $v$ | 7.4e-2 (2e-2) | 9.7e-2 (7e-3) | **5.5e-3** (8e-5) | 0.9559 (3e-2) | 0.9322 (1e-2) | **0.9995** (1e-5) |
| $p$ | 1.3e-1 (7e-2) | 1.1e-1 (2e-2) | **4.4e-2** (1e-2) | 0.5691 (1e-1) | 0.2626 (1e-1) | **0.7785** (9e-2) |
| $\sigma_{11}$ | 1.0e-1 (4e-2) | 2.8e-2 (3e-3) | **3.7e-3** (1e-4) | 0.9304 (4e-2) | 0.8693 (3e-2) | **0.9999** (2e-6) |
| $\sigma_{12}$ | 4.5e-2 (4e-3) | 7.5e-3 (4e-4) | **2.4e-3** (2e-5) | 0.9617 (5e-3) | 0.9868 (1e-3) | **0.9999** (1e-6) |
| $\sigma_{22}$ | 3.3e-2 (3e-3) | 2.7e-2 (1e-3) | **1.4e-3** (1e-5) | 0.9662 (6e-3) | 0.9782 (2e-3) | **0.9999** (1e-6) |
| AirfRANS dataset | | | | | | |
| $C_D$ | 6.1e-2 (2e-2) | 4.9e-2 (7e-3) | **3.3e-2** (2e-3) | 0.9596 (2e-2) | 0.9743 (1e-2) | **0.9831** (2e-3) |
| $C_L$ | 4.1e-1 (1e-1) | 2.4e-1 (8e-2) | **8.0e-3** (6e-4) | 0.9776 (8e-3) | 0.9851 (1e-2) | **0.9999** (2e-6) |
| $u$ | 5.6e-2 (3e-3) | 8.3e-2 (2e-3) | **1.8e-2** (9e-5) | 0.9659 (3e-3) | 0.9110 (3e-3) | **0.9749** (8e-5) |
| $v$ | 4.2e-2 (2e-3) | 1.2e-1 (2e-3) | **1.5e-2** (3e-5) | 0.9683 (3e-3) | 0.7516 (5e-3) | **0.9806** (3e-5) |
| $p$ | 8.5e-2 (7e-3) | 9.9e-2 (1e-2) | **5.1e-2** (2e-5) | 0.9602 (8e-3) | 0.9390 (2e-2) | **0.9934** (1e-5) |

**Uncertainty estimates.** Once trained, the MMGP model provides access to predictive uncertainties for the output fields and scalars. Figure 4 shows an example of predicted pressure field for an

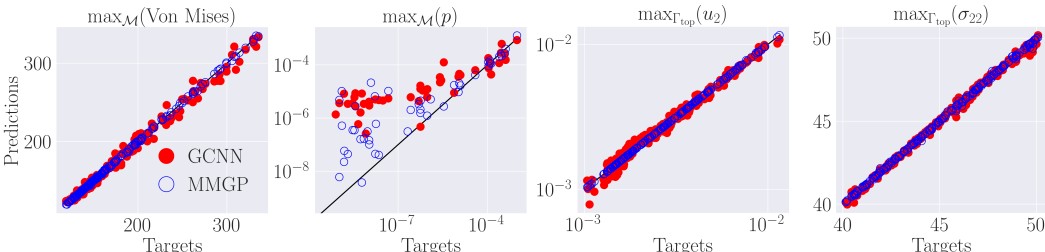

Figure 2: (`Tensile2d`) Test predictions versus test targets obtained for the output scalars of interest.

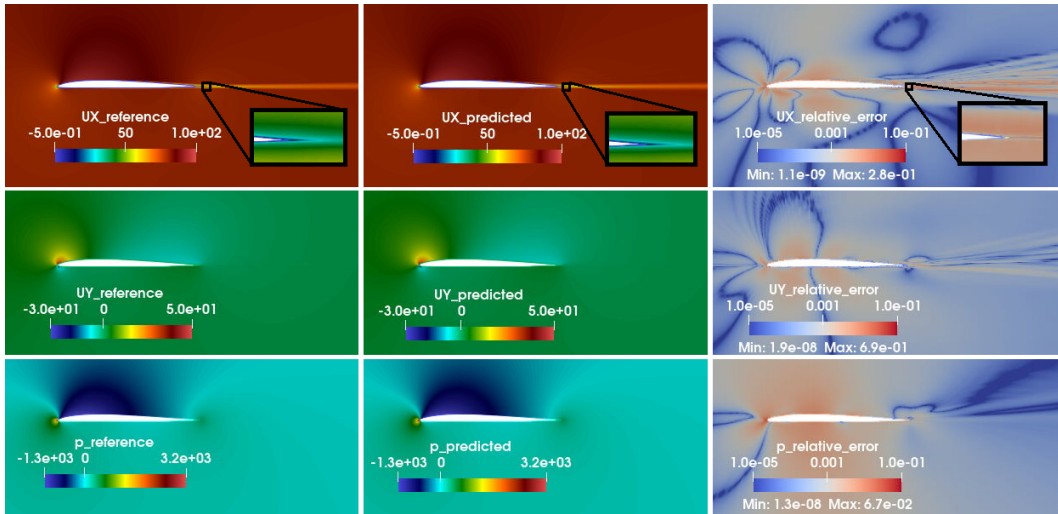

Figure 3: (`AirfRANS`) Test sample 787, fields of interest $u$ ($UX$), $v$ ($UY$) and $p$: (left) reference, (middle) MMGP prediction, (right) relative error.

arbitrary test input mesh of the `Rotor37` experiment, together with the predictive variance and the point-wise relative absolute error. High relative errors are localized where the pressure field exhibits a discontinuity, known as a shock in compressor aerodynamics. The predictive variance is also higher near this region, reflecting that the GP-based surrogate model is uncertain about its prediction of the shock position. Figure 5 shows graphs of two output scalars with respect to the input pressure in the `Tensile2d` problem. The predictive intervals of the MMGP model are discriminative: they get wider as the input pressure falls out of the support of the training distribution. In order to assess the validity of the prediction intervals, we compute the prediction interval coverage probability (PICP), *i.e.* the average of test targets that fall into the 95% prediction interval. For the `AirfRANS` dataset, PICPs of 93.05% and 93.5% for respectively the outputs $C_L$ and $C_D$ are obtained by averaging the individual PICPs of 10 independent MMGP models. The prediction intervals are slightly over-confident but this could be corrected by *e.g.* conformalizing the Gaussian process [69].

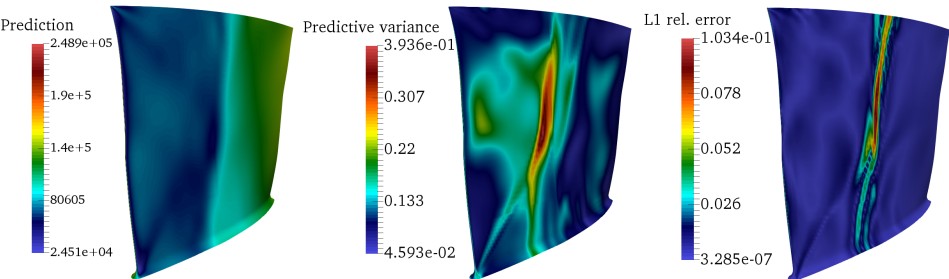

Figure 4: (`Rotor37`) MMGP: prediction, predictive variance, and $L^1$ relative error of the pressure field for an arbitrary geometry in the test dataset.

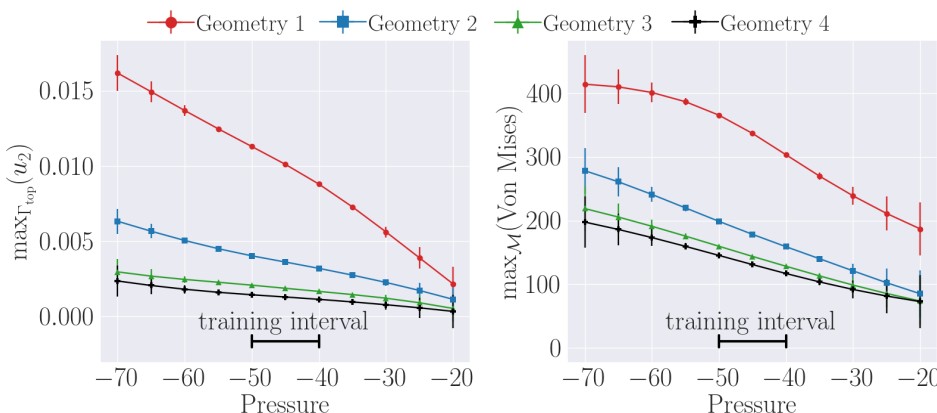

Figure 5: (`Tensile2d`) MMGP: graphs of the predicted $v_{\max}$ and $\sigma_v^{\max}$ with respect to the pressure, for four different test input meshes, and 11 values of input pressure that go beyond the training range $(-50, -40)$, with 95% confidence intervals.

**Computational times.** The MMGP model can easily be trained on CPU hardware and with much lower computational times, see Table 3.

Table 3: Training computational times: GCNN and MGN on a Nvidia A100 Tensor Core GPU (neural network training), MMGP on a 48 cores Intel Xeon Gold 6342 CPU (Gaussian process regressors training). Between parenthesis are indicated the numbers of trainings carried-out to optimize hyperparameters (best is **bold**).

| Dataset | GCNN | MGN | MMGP |
|---|---|---|---|
| `Rotor37` | $(200 \times)$ 24 h | $(6 \times)$ 13 h 14 min | $(10 \times)$ **2 min 49 s** |
| `Tensile2d` | $(200 \times)$ 1 h 25 min | $(6 \times)$ 6 h 50 min | $(10 \times)$ **1 min 38 s** |
| `AirfRANS` | $(200 \times)$ 5 h 15 min | $(6 \times)$ 5 h 00 min | $(10 \times)$ **5 min 47 s** |

## 5 Conclusion

In summary, our proposed method presents an innovative approach to approximating field and scalar quantities of interest within the context of solving complex physics problems for design optimization. Our work introduces two key contributions: firstly, the utilization of mesh morphing pretreatment in conjunction with finite element interpolation, and secondly, the incorporation of shape embedding through dimensional reduction of coordinates, treating them as continuous fields over the geometric support. These innovations alleviate the machine learning task from the challenges of handling variable-sized samples and the need to learn implicit shape embedding. By reducing the dimensionality of inputs and outputs, our approach allows for the application of efficient Gaussian process regression. Notably, our MMGP model exhibits several key advantages. It can seamlessly handle very large meshes, is amenable to efficient CPU training, is fairly interpretable, demonstrates high accuracy in our experimental results, and provides readily available predictive uncertainties.

Future works will explore the extension of our method to accommodate time-dependent quantities of interest and investigate the optimization of the morphing process to enhance data compression and overall performance. Our research opens exciting avenues for advancing the capabilities of machine learning in the realm of physics-based design optimization.

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

# Appendix

**Available dataset and code**   The code corresponding to the two-dimensional solid mechanics case (`Tensile2d`) described in Section 4.1 is available at `https://gitlab.com/drti/mmgp` [3]. A documentation is available at `https://mmgp.readthedocs.io/` [2], where details are provided on how to download the dataset `Tensile2d` and reproduce the corresponding numerical experiments.

Details regarding the datasets are provided in Appendix A. Morphing strategies and dimensionality reduction techniques are described in Appendices B and C. Details about the GNNs baselines are given in Appendix D. Finally, additional results about the considered experiments are gathered in Appendix E.

## A   Datasets

This section provides additional details regarding the synthetic datasets `Tensile2d` and `Rotor37`. Regarding the `AirfRANS` dataset, the reader is referred to [14].

### A.1   Rotor37 dataset

Examples of input geometries are shown in Figure 6 together with the associated output pressure fields. While the geometrical variabilities are moderate, it can be seen that they have a significant impact on the output pressure field. A design of experiment for the input parameters of this problem are generated with maximum projection LHS method [43]. For each input mesh and set of input parameters, a three-dimensional aerodynamics problem is solved with RANS, as illustrated in, *e.g.* [8]. The output scalars of the problem are obtained by post-processing the three-dimensional velocity.

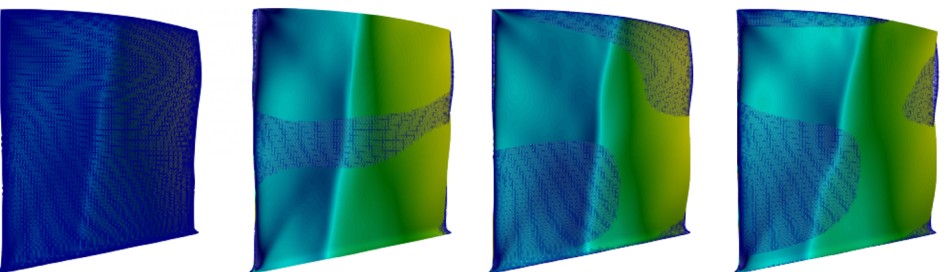

Figure 6: (`Rotor37`) Four geometries with their corresponding output pressure fields. The first panel shows the mesh, and the second to last panels show a superposition of the corresponding geometry and the mesh of the first one.

### A.2   Tensile2d dataset

Examples of input geometries are shown in Figure 7. A two-dimensional boundary value problem in solid mechanics is considered, under the assumption of small perturbations (see, *e.g.* [12]). The partial differential equation is supplemented with Dirichlet and Neumann boundary conditions: the displacement on the lower boundary is fixed, while a uniform pressure is applied at the top. The input parameters of the problem are chosen to be the magnitude of the applied pressure and 5 parameters involved in the elasto-visco-plastic constitutive law of the material [50]. The outputs of the problem are chosen as the components of the displacement field, $u$ and $v$, the entries of the Cauchy stress tensor, $\sigma_{11}$, $\sigma_{22}$, $\sigma_{12}$, and the cumulative plastic strain $p$. We also consider 4 output scalars obtained by post-processing the fields of interest: the maximum plastic strain $p_{\max}$ accross the geometry, the maximum vertical displacement $v_{\max}$ at the top of the geometry, and the maximum normal stress $\sigma_{22}^{\max}$ and Von Mises stress $\sigma_v^{\max}$ accross the geometry. It is worth emphasizing that the cumulative plastic strain $p$ is challenging to predict, as illustrated in Section E.

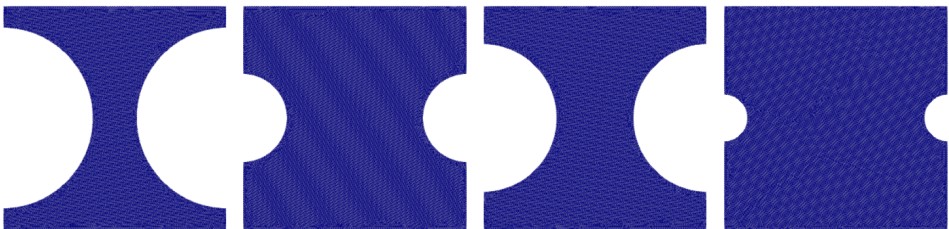

Figure 7: (`Tensile2d`) Illustration of the four input meshes that are used in Figure 5.

# B  Morphing strategies

In this section, we briefly describe the Tutte's barycentric mapping [71] and the radial basis function morphing [9, 21] used in the considered experiments.

**Tutte's barycentric mapping.**    For this method, we are limited to connected triangular surface meshes of fixed topology, either in a 2D or in a 3D ambient space. Tutte's barycentric mapping starts by setting the value of the displacement of the boundary points of the mesh (usually onto the unit disk), and solve for the value at all the remaining nodes of the mesh. The physical features available on the mesh, and inherited from the problem, are used in the specification of the displacement of the boundary nodes.

We recall that $\mathbf{x}_I$, $I = 1\ldots N$, denote the mesh nodes coordinates. We assume that the numbering of the nodes starts with the interior points of the mesh $1\ldots N_{\text{int}}$, and ends with the $N_b$ nodes on its boundary $N_{\text{int}} + 1\ldots N$. The morphed mesh node coordinates are denoted by $\overline{\mathbf{x}}_I$, $I = 1\ldots N$. The coordinates of the boundary of the morphed mesh being known, we denote $\overline{\mathbf{x}}_{b_I} = \overline{\mathbf{x}}_{I+N_{\text{int}}}$, $I = 1\ldots N_b$. Then the following sparse linear system is solved for the morphing of the interior points:

$$\overline{\mathbf{x}}_I - \frac{1}{d(I)} \sum_{J \in \mathcal{N}(I) \cap [\![1, N_{\text{int}}]\!]} \overline{\mathbf{x}}_J = -\frac{1}{d(I)} \sum_{J \in \mathcal{N}(I) \cap [\![N_{\text{int}}, N]\!]} \overline{\mathbf{x}}_{b_{J-N_{\text{int}}}},$$

where $\mathcal{N}(I)$ and $d(I)$ are respectively the neighbors and the number of neighbors of the node $I$ in the graph (or in the mesh following its connectivity).

In the 2D solid mechanics case `Tensile2d`, we know the rank of the point separating the left and the bottom faces which we map onto the point $(0, 1)$ of the target unit disk. The linear density of nodes on the boundary of the target unit disk is chosen to be the same as the one of the mesh sample (relative to the length of the boundary), see Figure 8 for an illustration.

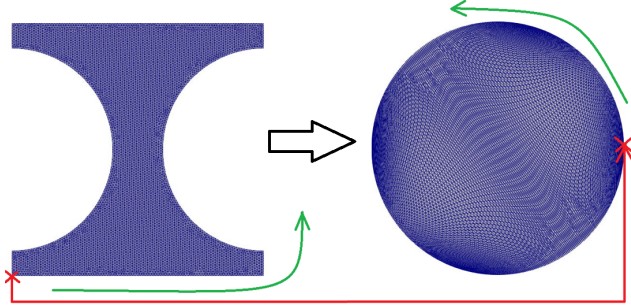

Figure 8: (`Tensile2d`) Illustration of the Tutte's barycentric mapping used in the morphing stage.

From [27, corollary 2], the morphing described above is called a parametrization, and defines an isomorphic deterministic transformation of the considered triangular surface mesh $\mathcal{M}$, into a plane triangular mesh $\overline{\mathcal{M}}$ of the unit disk. Notice that although these morphing techniques are called "mesh parametrization", this do not mean that we need to know any parametrization of the shape: these are deterministic transformations of the meshes, requiring no other information than the nodes locations and the triangles connectivities.

This method is taken from the computer graphics community and has been improved over the years. In [78], a quality indicator called stretch metric is optimized during an iterative procedure, to obtain more regular morphed mesh. Recently in [31], a procedure was proposed to drastically improve mesh parametrization, even in difficult cases where some triangles are overlapping. It should be noted that such iterative procedures come with the additional cost of solving a series of sparse linear systems.

**Radial Basis Function morphing.** In the same fashion as Tutte's barycentric mapping, RBF morphing methods start by setting the value of the displacement at some particular nodes of the mesh (here the boundary points of the mesh, but interior points can be considered as well with RBF), and solve for the location at all the remaining nodes of the mesh. The physical features available on the mesh are also used in the specification of the displacement of the boundary points. RBF morphing methods are compatible with 2D and 3D structured and unstructured meshes, do not require any mesh connectivity information, and can be easily implemented in parallel for partitioned meshes.

We use the RBF morphing method as proposed in [21]. Once the mapping for the $N_b$ boundary points of ranks $N_{\text{int}} + 1 \ldots N$ is fixed, then the interior points $1 \leq I \leq N_{\text{int}}$ are mapped such as

$$\overline{\mathbf{x}}_I = \sum_{J=1}^{N_b} \alpha_J \phi(\|\mathbf{x}_I - \mathbf{x}_{b_J}\|), \quad 1 \leq I \leq N_{\text{int}},$$

where $\phi$ is a radial basis function with compact support and $\alpha_J$ are determined by the interpolation conditions. More precisely, we choose the radial basis function with compact support $\phi(\xi) = (1 - \xi)^4 (4\xi + 1)$ and support radius equal to half of the mesh diameter, and interpolation conditions means that the morphing is known at the boundary points:

$$\mathbf{M}_{\text{RBF}} \alpha = \overline{\mathbf{x}}_b,$$

where $\mathbf{M}_{\text{RBF}_{I,J}} = \phi(\|\mathbf{x}_{b_I} - \mathbf{x}_{b_J}\|), 1 \leq I, J \leq N_b$.

For the `AirfRANS` dataset, we make use of the physical properties of the boundary condition to morph each mesh onto the first geometry of the training set. Referring to Figure 9 (bottom), we know which nodes lie the external boundary (in black), airfoil extrado (in red), airfoil intrado (in blue) and which nodes define the leading and trailing points (green crosses). We choose to keep the points at the external boundary fixed (zero mapping), map the leading and trailing edge to the ones of the mesh of the first training sample, and map the points on the extrado and intrado along the ones of the mesh of the first training sample while conserving local node density (relative to the length of the boundary). A zoom of the RBF morphing close to the airfoil for test sample 787 is illustrated in Figure 10.

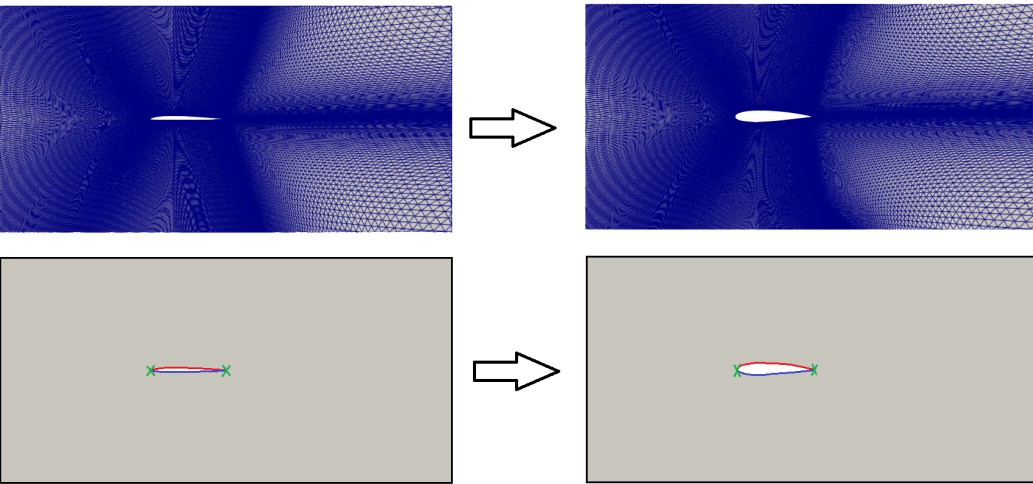

Figure 9: (`AirfRANS`) RBF morphing for test sample 787; (top) complete mesh morphing, (bottom) illustration of the mapping of the boundary points.

Notice that while Tutte's barycentric mapping requires solving a sparse linear system of rather large size $N_{\text{int}}$, RBF morphing requires solving a dense linear system of smaller size $N_b$. RBF morphing methods dealing with complex non-homogeneous domains have been proposed in [15].

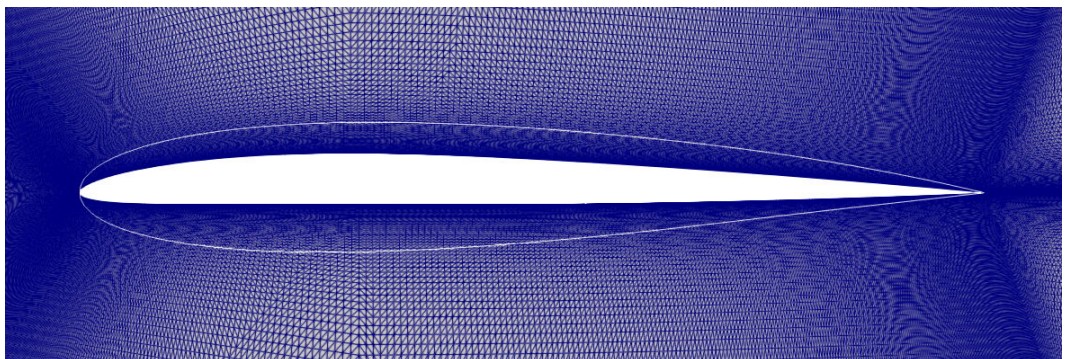

Figure 10: (AirfRANS) Zoom of the RBF morphing close to the airfoil for test sample 787.

**Other methods.** In [23], the morphing is computed by means of solving an elastic problem. See also [6, 70] for literature reviews on mesh morphing methods. Mesh deformation algorithms compatible with topology changes have been proposed [80].

## C   Dimensionality reduction

The principal component analysis can be replaced by more effective dimensionality reduction techniques such as the snapshot-POD. The latter is a variant where the underlying $\ell^2$-scalar product used to compute the coefficients of the empirical covariance matrices is replaced by the $L^2(\overline{\mathcal{M}}^c)$-inner product. Define the symmetric positive-definite matrix $\mathbf{M} \in \mathbb{R}^{N_c \times N_c}$, such that

$$M_{IJ} = \int_{\overline{\mathcal{M}}^c} \overline{\varphi}_I^c(\overline{\mathbf{x}}) \overline{\varphi}_J^c(\overline{\mathbf{x}}) d\overline{\mathbf{x}}.$$

In general, a quadrature formula, in the form of a weighted sum over function evaluations on the common mesh, is chosen such that the integral are computed exactly for functions in the span of the finite element basis. Then, the empirical covariance matrix is computed as

$$\left( (\widetilde{\mathbf{U}}_k^i)^T \mathbf{M} \widetilde{\mathbf{U}}_k^j \right)_{i,j} = \sum_{I,J=1}^{N_c} \int_{\overline{\mathcal{M}}^c} \overline{\mathcal{U}}_k^i(\overline{\mathbf{x}}_I^c) \overline{\varphi}_I^c(\overline{\mathbf{x}}) \overline{\mathcal{U}}_k^j(\overline{\mathbf{x}}_J^c) \overline{\varphi}_J^c(\overline{\mathbf{x}}) d\overline{\mathbf{x}} = \int_{\overline{\mathcal{M}}^c} P(\overline{\mathcal{U}}_k^i)(\overline{\mathbf{x}}) P(\overline{\mathcal{U}}_k^j)(\overline{\mathbf{x}}) d\overline{\mathbf{x}},$$

which corresponds to the continuous formula for the computation of the correlations of the fields of interest transported and interpolated on the common morphed mesh. Hence, the empirical covariance matrix can take into account any heterogeneity of the common morphed mesh, which may occur after morphing. The same construction can be made for the spatial coordinate field, while its derivation is more technical, because it involves vector fields instead of scalar fields. The computation of the empirical covariance matrix can be easily be parallelized on numerous computer nodes, provided that the common morphed mesh has been partitioned in subdomains, which enable efficient dimensionality reduction for meshes up to millions of degrees of freedom, see [20].

Other linear or nonlinear dimension reduction techniques can be considered, like mRMR feature selection [22, 62], kernel-PCA [68] or neural network-based autoencoders [48].

## D   Architectures and hyperparameters of GNN-based baselines

### D.1   Graph convolutional neural network

A graph convolutional neural network (GCNN) [67] has been implemented using `PyTorch Geometric` [26] with the Graph U-Net [29] architecture and the following specifications: (i) $top_k$ pooling [29, 47] layers with a pooling ratio of $0.5$ to progressively aggregate information over nodes of the graph, (ii) feature sizes progressively increased after each $top_k$ pooling, *i.e.*, 16, 32, 64, 96 and 128, (iii) between each pooling, residual convolution blocks [40] are added to combine two consecutive normalization-activation-convolution layers, (iv) BatchNorm [42] layers are introduced, and (v) LeakyReLU [57] activations are used with slope of $0.1$ on negative values.

A weighted multi-loss $\mathcal{L}$ that combines scalars and fields is used, and defined as

$$\mathcal{L}\left((\mathbf{U}, \mathbf{w}), (\mathbf{U}', \mathbf{w}')\right) = \lambda_{\mathrm{scalars}}\mathcal{L}_{\mathrm{MSE}}\left(\mathbf{w}, \mathbf{w}'\right) + \lambda_{\mathrm{fields}} \sum_{k=1}^{d} \mathcal{L}_{\mathrm{MSE}}\left(\mathbf{U}_k, \mathbf{U}'_k\right),$$

where $\lambda_{\mathrm{scalars}}$ and $\lambda_{\mathrm{fields}}$ are two positive hyperparameters. For gradient descent, an Adam optimizer [45] is used with a cosine-annealing learning rate scheduler [55]. The following hyperparameters are optimized by grid search: (i) the learning rate, 13 values between $1.0$ and $0.0001$, (ii) the weight $\lambda_{\mathrm{field}} \in \{1, 10, 100, 1000\}$, and (iv) the type of convolution, chosen between *GATConv* [72], *GeneralConv* [79], *ResGatedGraphConv* [16] and *SGConv* [77]. There are many other hyperparameters that could be tuned, as the number of layers or the number of features on each layer. The chosen hyperparameters are summarized in Table 4 for each experiment. In the case of the `Rotor37` problem,

Table 4: Chosen hyperparameters for the GCNN architectures.

| Dataset | Learning rate | $\lambda_{\mathrm{field}}$ | Convolution |
|---|---|---|---|
| Rotor37 | 0.02 | 10.0 | GeneralConv |
| Tensile2d | 0.01 | 100.0 | GeneralConv |
| AirfRANS | 0.005 | 10.0 | GeneralConv |

the outwards normals to the surface of the compressor blade are added as input features to input graphs. Similarly, for the `Tensile2d` and `AirfRANS` problems, the signed distance function is added as an input feature.

### D.2  MeshGraphNets

The MGN model [63] is taken from Nvidia's `Modulus` [4] package that implements various deep surrogate models for physics-based simulations. The same set of hyperparameters is used for all the considered regression problems, which is chosen after conducting a grid search over the learning rate, the number of hidden nodes and edges features, the number of processor steps. The learning rate is set to $0.001$, the numbers of hidden features `hidden_dim_node_encoder`, `hidden_dim_edge_encoder`, and `hidden_dim_node_decoder` are all set to $16$. The number of processor steps is chosen as $10$. The rest of the MGNs hyperparameters are left to the default values used in the `Modulus` package. The batch size is set to $1$, the activation is chosen as the LeakyReLU activation with a $0.05$ slope, and $1,000$ epochs are performed for training the network. For scalar outputs, a readout layer taken from [46] is added to the model. The input nodes features are given by the spatial coordinates of the nodes, and possible additional fields such as the signed distance function (for the `Tensile2d` and `AirfRANS` problems), or the outward normals (for the `Rotor37` problem). Given two node coordinates $\mathbf{x}_i$ and $\mathbf{x}_j$, the edges features are chosen as $\exp(-\|\mathbf{x}_i - \mathbf{x}_j\|_2^2/(2h^2))$, where $h$ denotes the median of the edge lengths in the mesh.

For each considered regression problem, it is found that it is more effective to train two MGNs models, one dedicated to handling output fields and the other specialized for output scalars, respectively. Nevertheless, better hyperparameter tuning and more effective readout layers could lead to different conclusions regarding this matter.

### D.3  Training on `AirfRANS`

As mentioned in Section 4.1, training GNNs on the `AirfRANS` dataset is computationally expensive due to the sizes of the input meshes. For this reason, the GNN-based baselines are trained on the `AirfRANS-remeshed` dataset (see Table 1) obtained by coarsening the input meshes (see Figure 11) and the associated output fields. Once trained, predictions on the initial fine meshes are obtained through finite element interpolation. It should be underlined that this strategy may hinder the performance of the GNN-based baselines, as the reconstructed fields are obtained by finite element interpolation.

## E  Additional results

This section gathers additional results about the experiments considered in Section 4.

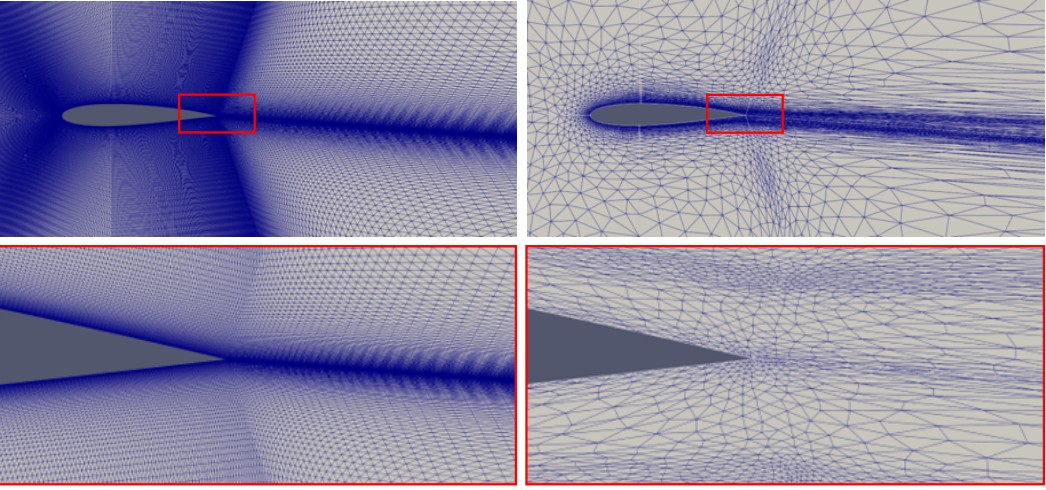

Figure 11: (`AirfRANS`) Example of an original mesh from the dataset (left) and the corresponding coarsened mesh in the `AirfRANS-remeshed` dataset (right).

### E.1 Out-of-distribution inputs

Figures 12, 13, and 14 show histograms of the logarithm of the predictive variance for the output scalars of interest on different sets of samples. The aim is to empirically assess if the MMGP model is able to identify out-of-distribution (OOD) inputs by attributing higher predictive variances. In the case of the `Rotor37` problem, three OODs samples are generated such that the support of the covariates ($\mu_1$, $\mu_2$, and $\mathcal{M}$) are disjoint with the support of the training distributions. It can be seen that the variances of the OOD samples are higher than the ones of the in-distribution samples. Similar observations are made for the `Tensile2d` and `AirfRANS` problems. While such an analysis can help to identify OOD inputs, it should be underlined that the predictive uncertainties of Gaussian processes are only valid under the Gaussian a priori assumption, which may not be verified in practice. For instance, the ellipsoid geometry has a similar variance as the in-distribution samples in Figure 13.

### E.2 Predicted output fields

**Tensile2d dataset.** For reproducibility matters, we mention that for the field $p$ and the scalar $p_{\max}$, the denominators in the formulae $\mathrm{RRMSE}_f$ and $\mathrm{RRMSE}_s$ has been replaced by 1 when its value is below $1e-6$ for preventing division by zero, which corresponds to replacing the relative error by the absolute error for samples that do not feature plastic behaviors.

In Figures 15-18, we illustrate the MMGP prediction, variance and relative error for all the considered fields: $u$, $v$, $p$ (evrcum), $\sigma_{11}$, $\sigma_{12}$ and $\sigma_{22}$, for respectively the first training inputs, first test inputs, and two out-of-distribution geometries (ellipsoid and wedge). In particular, the wedge cut-off geometry features stress concentrations that are not present in the training set. We notice that the predictions for selected train and test inputs (Figures 15 and 16) are accurate, with small relative errors and relatively small predictive variances, except for some small areas where the considered fields have larger magnitudes. As expected, the predictions for the ellipsoid and wedge cases (Figures 17 and 18) are less accurate than for in-distribution shapes, but the predictive variances are larger, which confirms that MMGP informs that, locally, the prediction cannot be trusted. This phenomenon is particularly strong for the wedge case, that largely differs from the training set shapes.

In Figures 19 and 20, we consider all the output interest fields. The 2D domain is visualized in 3D, in the form of three surfaces: a transparent blue for the 0.025-quantile, a white for the reference prediction and a transparent red for the 0.975-quantile. The point-wise $95\%$ confidence interval is the distance (along the out-of-plane axis) between the transparent blue and red surfaces. We notice that the $95\%$ confidence intervals are very small for the train and test inputs, larger for the ellipsoid case, and much larger for the wedge case (in particular $\sigma_{22}$). Not surprisingly, for the OOD shapes

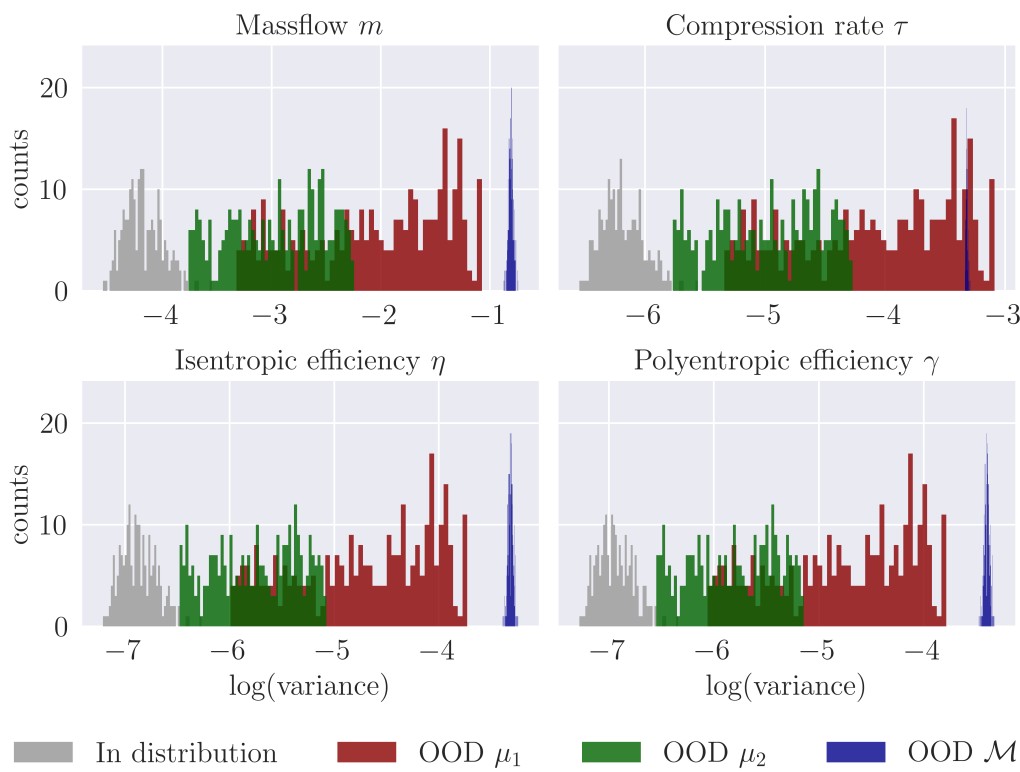

Figure 12: (Rotor37) Histograms of log(variance) of MMGP predictions for the output scalars of interest on four sets: in grey the testing set (in distribution), in green and red respectively two sets where the input pressure $\mu_1$ and rotation speed $\mu_2$ are taken OOD, and in blue a set of geometry taken OOD.

ellipsoid and wedge, some surfaces intersect, meaning that, locally, the reference solution is not inside the 95% confidence interval.

In Figure 21, we illustrate the finite element error occurring when predicting $\sigma_{11}$ with respect to the 95% confidence interval for samples taken from the training and testing sets. We notice that on the training set, the finite element error magnitude is comparable to the 95% confidence interval, which is very small on training samples. On the testing set, the 95% confidence interval is larger, and the finite element error magnitude can be neglected.

**AirfRANS dataset.** Figures 22 and 23 illustrate reference, MMGP prediction and relative errors for the fields of interest $u$, $v$ and $p$ on respectively test sample 430 and train sample 93. In the first row, zooms are provided close to the trailing edge to illustrate the accuracy of the prediction in the thin boundary layer. Relative errors have larger magnitudes on spatially restricted areas. We notice that on train sample 93, the areas with low relative error are larger than for test sample 430.

In Table 5, we compare MMGP and our trained GCNN and MGN, as well as the four models trained in [14], for the scalars of interest drag coefficient $C_D$ and lift coefficient $C_L$, computed by post-processing the predicted fields instead of directly predicting them as output scalars. This post-processing consists in integrating the reference and predicted wall shear stress (from the velocity) and pressure fields around the surface of the airfoil. The models from [14] are a MLP (a classical Multi-Layer Perceptron), a GraphSAGE [38], a PointNet [64] and a Graph U-Net [29] and the corresponding results are taken from [14, Table 19] ("full dataset" setting that we considered in this work). Refer to [14, appendix L] for a description of the used architecture. The limits of this comparison are (i) the mesh supporting the fields are not the same (they have been coarsen in [14] by process different from ours), (ii) the scalar integration routine are not identical (we integrate using

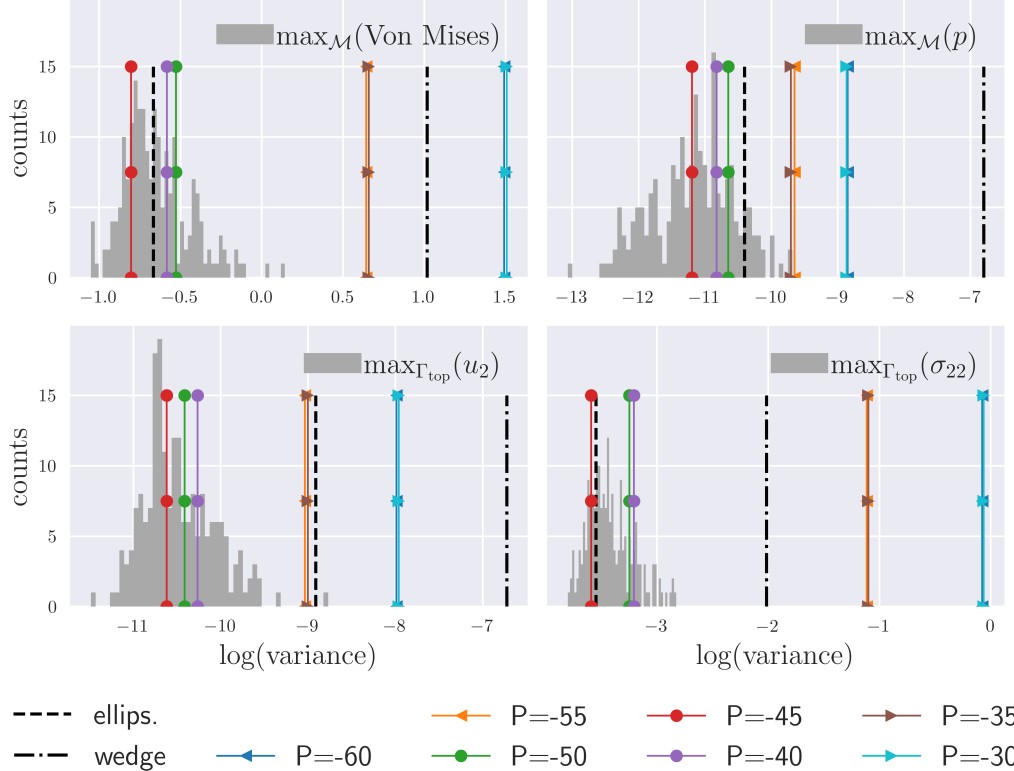

Figure 13: (`Tensile2d`) Histograms of log(variance) of MMGP predictions in grey for the output scalars of interest on the testing set (in distribution). The variance of the MMGP prediction is identified for various configurations: the ellipsoid and wedge cases (where all the nongeometrical parameters are taken at the center of the training intervals), and 5 settings where the same geometry is taken in the testing set and the input pressure varies (the training interval is $[-50, -40]$).

finite element representations). Within these limits, MMGP appears competitive with respects to the models of [14] and our trained GCNN and MGN.

Table 5: (`AirfRANS`) Relative errors (Spearman's rank correlation) for the predicted drag coefficient $C_D$ ($\rho_D$) and lift coefficient $C_L$ ($\rho_D$) for the four models of [14, Table 19], as well as GCNN, MGN and MMGP. These scalars of interest are computed as a postprocessing of the predicted fields (best is **bold**).

| Model | Relative error | | Spearman's correlation | |
| | $C_D$ | $C_L$ | $\rho_D$ | $\rho_L$ |
|---|---|---|---|---|
| MLP | 6.2e+0 (9e-1) | 2.1e-1 (3e-2) | 0.25 (9e-2) | 0.9932 (2e-3) |
| GraphSAGE | 7.4e+0 (1e+0) | 1.5e-1 (3e-2) | 0.19 (7e-2) | 0.9964 (7e-4) |
| PointNet | 1.7e+1 (1e+0) | 2.0e-1 (3e-2) | 0.07 (6e-2) | 0.9919 (2e-3) |
| Graph U-Net | 1.3e+1 (9e-1) | 1.7e-1 (2e-2) | 0.09 (5e-2) | 0.9949 (1e-3) |
| GCNN | 3.6e+0 (7e-1) | 2.5e-1 (4e-2) | 0.002 (2e-1) | 0.9773 (4e-3) |
| MGN | 3.3e+0 (6e-1) | 2.6e-1 (8e-2) | 0.04 (3e-1) | 0.9761 (5e-3) |
| MMGP | **7.6e-1** (4e-4) | **2.8e-2** (4e-5) | **0.71** (1e-4) | **0.9992** (2e-6) |

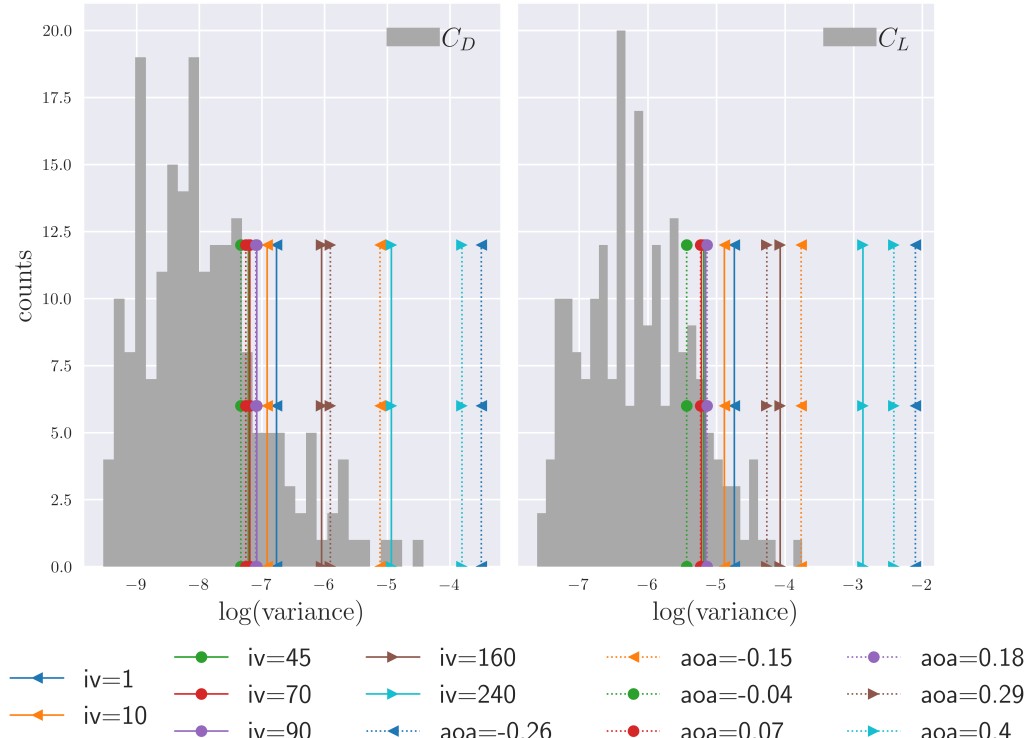

Figure 14: (`AirfRANS`) Histograms of log(variance) of MMGP predictions in grey for the output scalars of interest on the testing set (in distribution). The variance of the MMGP prediction is identified for various configurations, where the same geometry is taken in the testing set and the inlet velocity (iv) and angle of attack (aoa) varies (only iv=45, 70, 90 and aoa=-0.04, 0.07, 0.18 are in the training intervals).

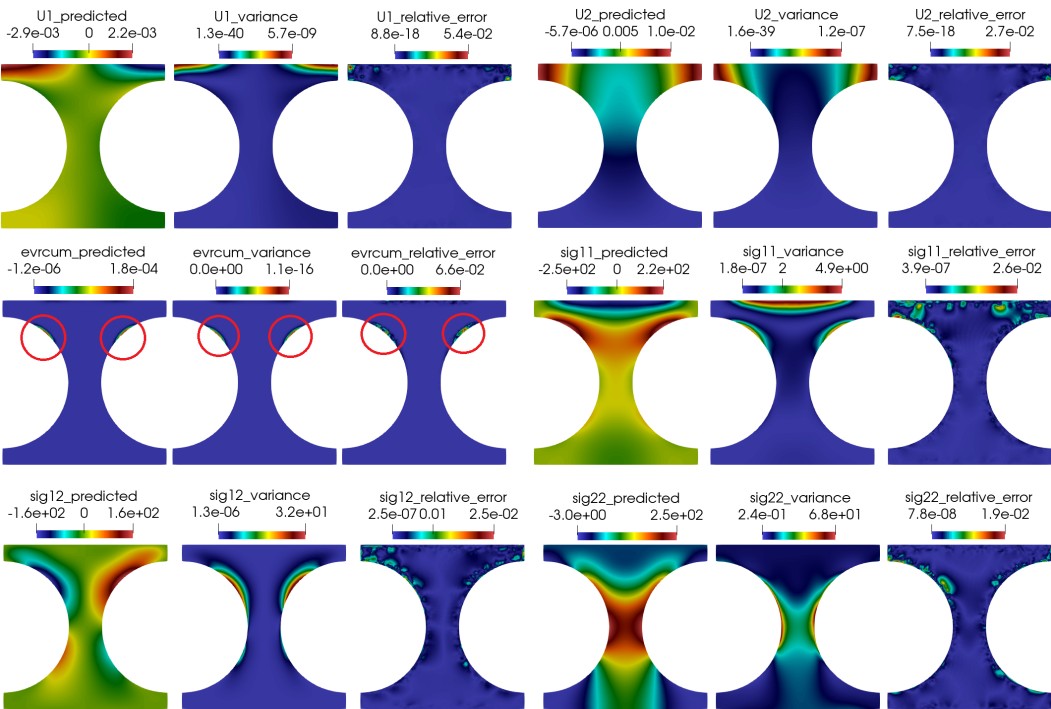

Figure 15: (`Tensile2d`) MMGP prediction for the first training input, where: $U_1$, $U_2$, evrcum, sig11, sig22, and sig12 correspond to $u$, $v$, $p$, $\sigma_{11}$, $\sigma_{22}$, and $\sigma_{12}$, respectively.

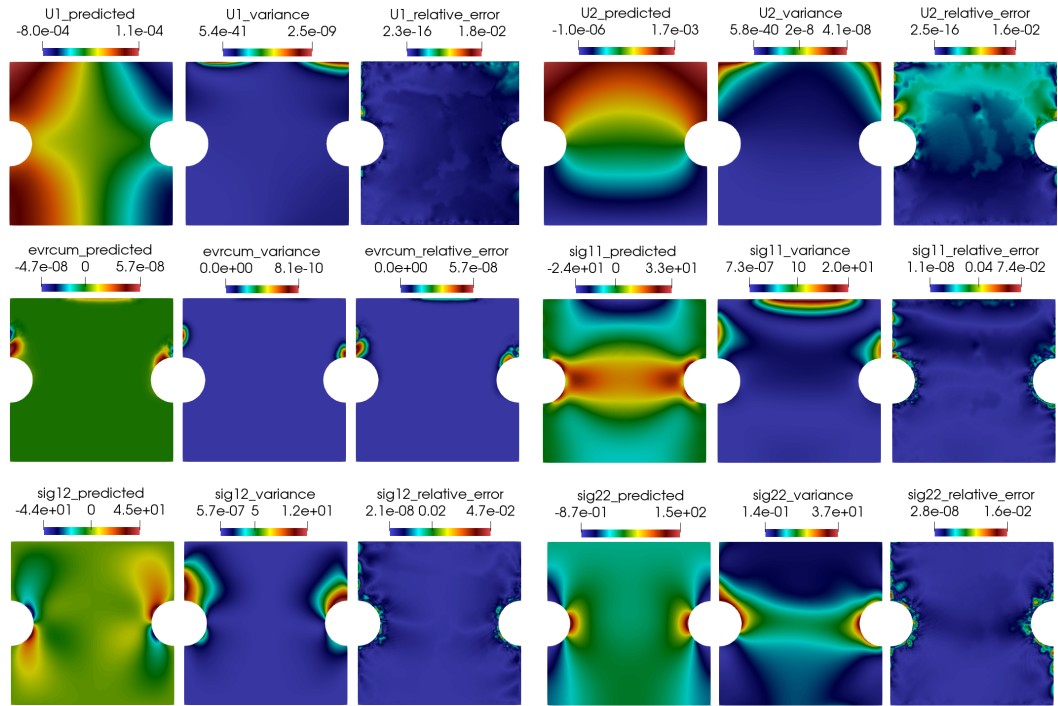

Figure 16: (Tensile2d) MMGP prediction for the first test input, where: $U_1$, $U_2$, evrcum, sig11, sig22, and sig12 correspond to $u$, $v$, $p$, $\sigma_{11}$, $\sigma_{22}$, and $\sigma_{12}$, respectively.

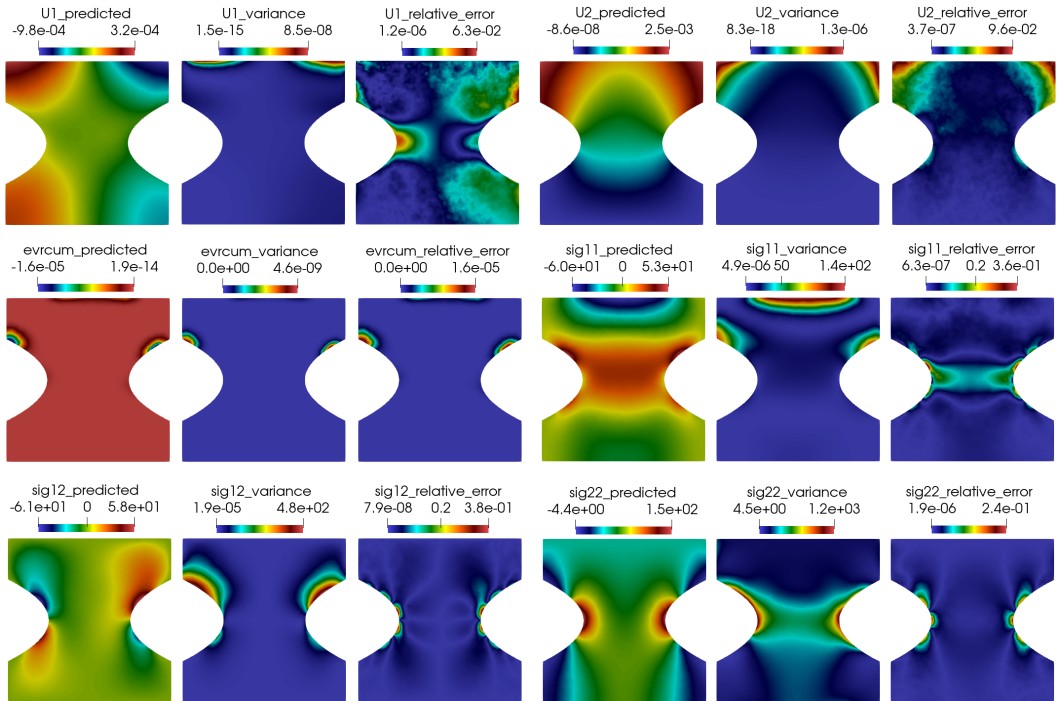

Figure 17: (Tensile2d) MMGP prediction for an OOD ellipsoid geometry, where: $U_1$, $U_2$, evrcum, sig11, sig22, and sig12 correspond to $u$, $v$, $p$, $\sigma_{11}$, $\sigma_{22}$, and $\sigma_{12}$, respectively.

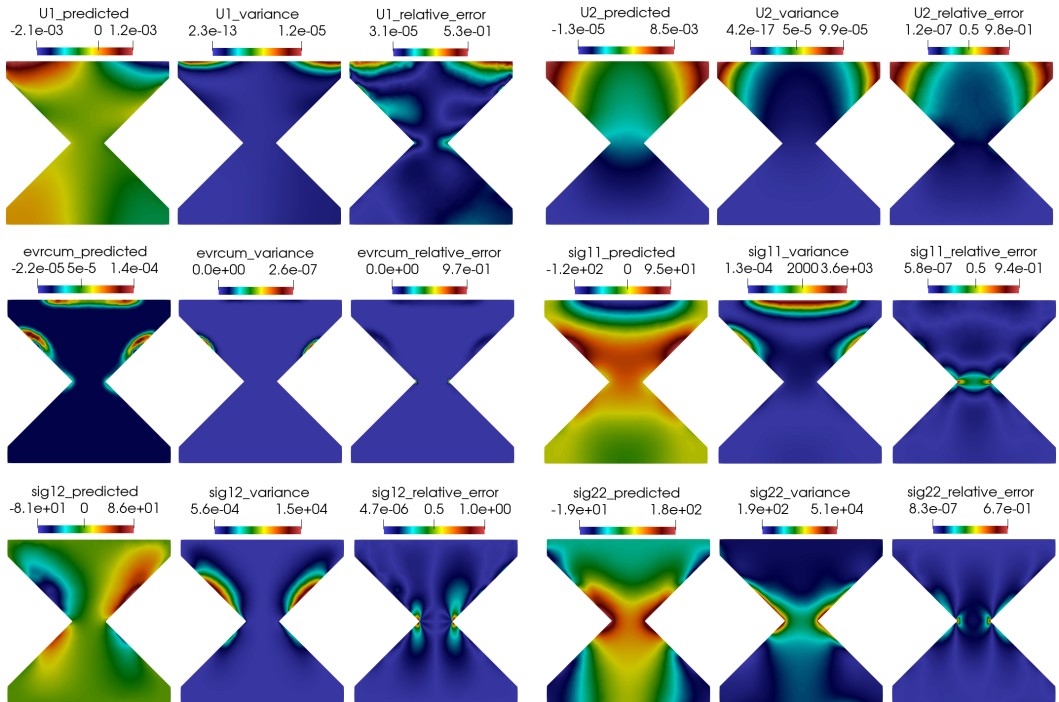

Figure 18: (Tensile2d) MMGP prediction for an OOD wedge cut-off geometry, where: $U_1$, $U_2$, evrcum, sig11, sig22, and sig12 correspond to $u$, $v$, $p$, $\sigma_{11}$, $\sigma_{22}$, and $\sigma_{12}$, respectively.

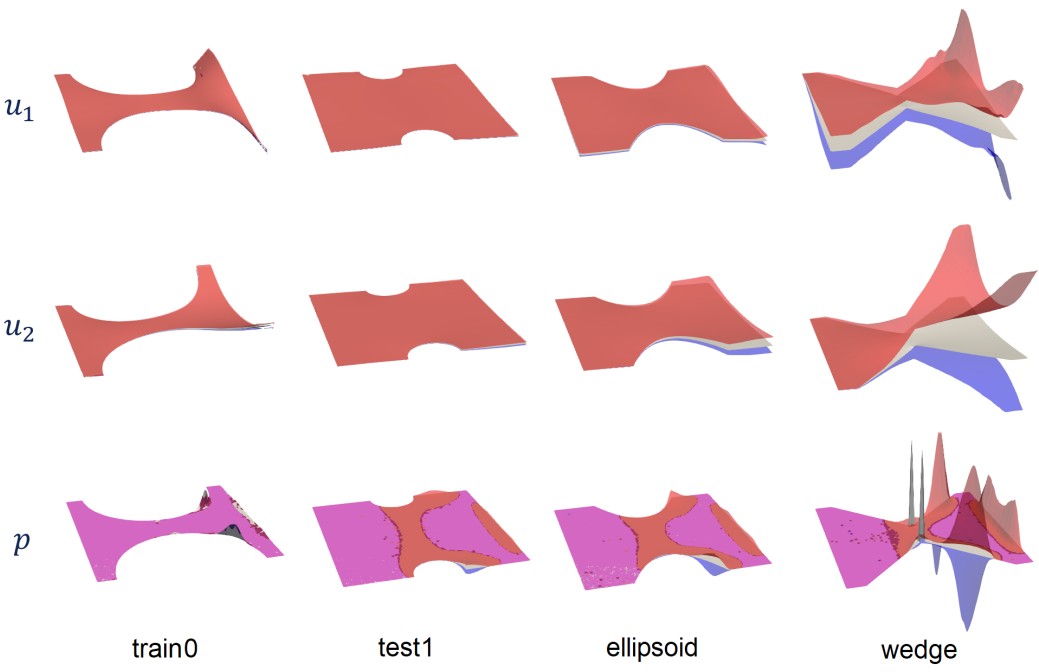

Figure 19: (Tensile2d) MMGP: train0, test1, ellipsoid and wedge cases, confidence intervals for $u$, $v$ and $p$ visualized as surfaces (for each field, the deformation factor is taken identical through the cases).

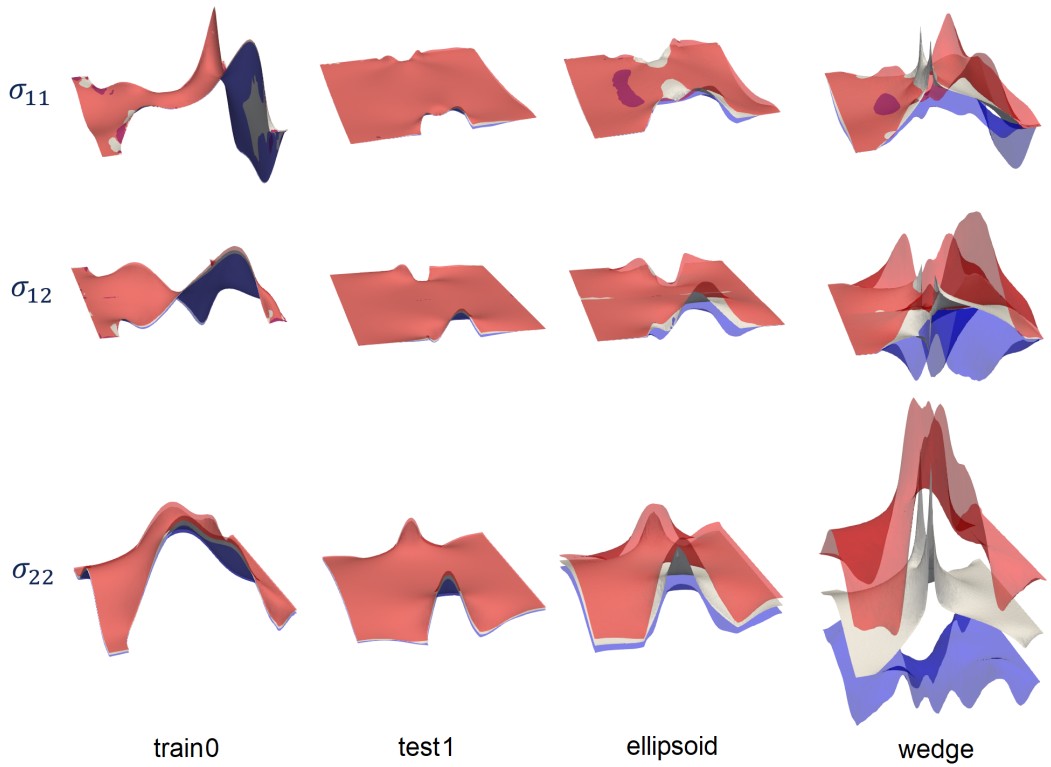

Figure 20: (Tensile2d) MMGP: train0, test1, ellipsoid and wedge cases, confidence intervals for $\sigma_{11}$, $\sigma_{12}$ and $\sigma_{22}$ visualized as surfaces (for each field, the deformation factor is taken identical through the cases).

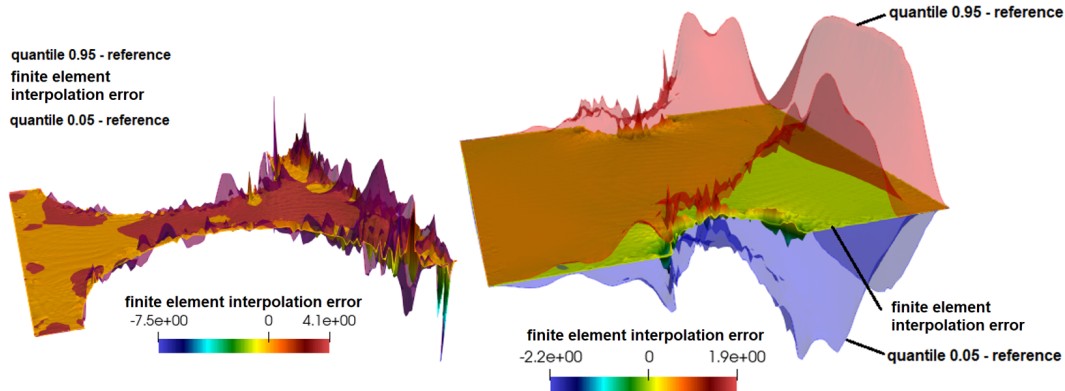

Figure 21: (Tensile2d) Finite element interpolation error for the prediction of $\sigma_{11}$ compared to the 95% confidence interval for a sample from (left): the training set, (right) the testing set.

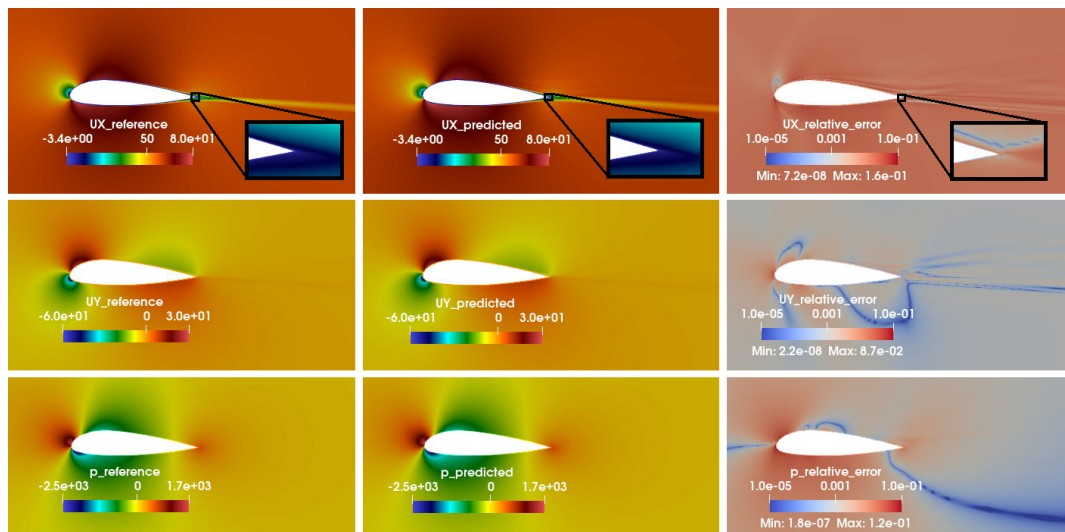

Figure 22: (`AirfRANS`) Test sample 430, fields of interest $u$ ($UX$), $v$ ($UY$) and $p$: (left) reference, (middle) MMGP prediction, (right) relative error.

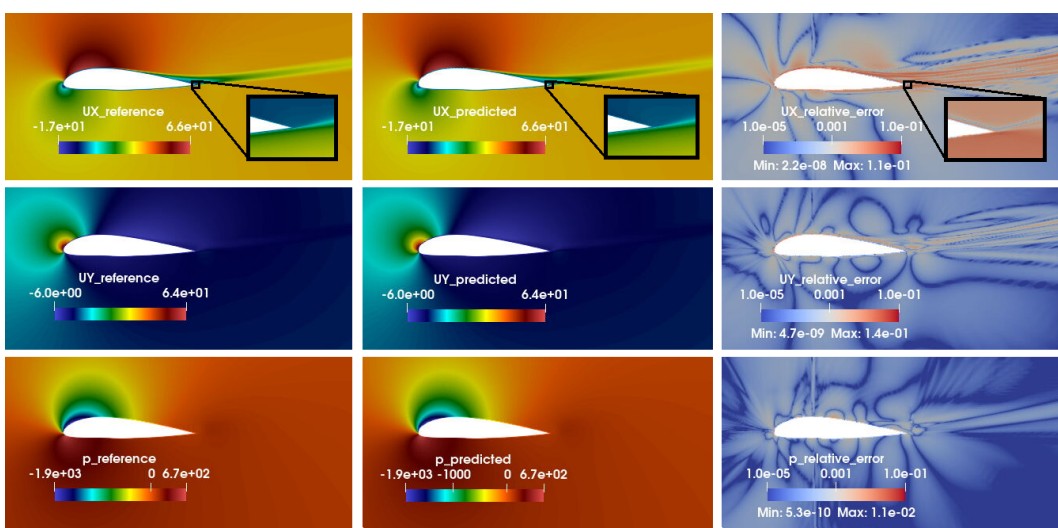

Figure 23: (`AirfRANS`) Train sample 93, fields of interest $u$ ($UX$), $v$ ($UY$) and $p$: (left) reference, (middle) MMGP prediction, (right) relative error.

