# OpenReview forum: "MMGP: a Mesh Morphing Gaussian Process-based machine learning method for regression of physical problems under nonparametrized geometrical variability"
_NeurIPS.cc/2023/Conference — NeurIPS 2023 poster_

### Official Review · Reviewer_4CEk · 2023-07-04

**Soundness:** 3 good
**Presentation:** 4 excellent
**Contribution:** 3 good
**Rating:** 7
**Confidence:** 4

**Summary:**

Authors present a GP-based method for predicting outcomes of a physical simulation under variations in geometry and physical parameters. The method relies on deterministic pre-processing of meshes and a GP on a reduced dimensional space. The pre-processing puts meshes of different structure, e.g., different number of nodes, on a common mesh so that function values on the meshes can be compared. One crucial point that allows this is to see coordinates as functions on the meshes. All function values on the common mesh coming from different meshes are fed into a PCA to reduce their dimensions. GP runs on the reduced dimensional representations.

**Strengths:**

1. The proposed method – to the best of my knowledge – is quite novel.
2. Leveraging mesh interpolation and morphing to help compare meshes of different structure is a great approach. Relying on deterministic operations reduce variability so that fewer examples are enough to learn powerful models. Combining these two is a great idea.
3. Use of GP instead of DL is another great aspect. Here, one can quantify the uncertainty in the predictions with ease and potentially identify out-of-distribution geometries or parameters. This leads to identification of cases where the model predictions are no longer reliable. I suggest authors to discuss this aspect as well and perhaps even do some experiments.
4. The article is very well written. This is surely a complicated method that requires background knowledge. Even in that setting, authors did a fantastic job in explaining the details.

**Weaknesses:**

1. ML-based methods for PDE systems currently only act as rough predictions. The proposed method is no exception. The predictions are not as accurate as simulations. For applications where accuracy is needed, current ML-based methods – including this work – can only act as a initialization. I believe this needs to be clearly stated.
2. In the introduction authors menthion the computational expense of traditional methods and their need for discretization. There are two aspects that needs to be mentioned.
    a) The proposed method also needs good discretization since it uses sample simulations, i.e., training set, in a GP framework. If those samples are bad, predictions will be bad.
    b) Traditional methods have guarantees over error bounds. This is a priceless aspect of these works. ML-based methods – including this one – does not provide any guarantees.
3. There are two parts that can be better explained:
        a) How do authors deal with arbitrary rigid body transformations? The technique seems to assume there are no rigid body transformations between different meshes.
        b) What is p on line 157?
4. Authors do not take into account Jacobian of coordinate transformations when morphing to the common mesh while computing new values of interest on the nodes. Ideally, such transformations should be taken into account when computing U values in Equation 4. It is true that authors do a backward transformation for the final predictions. However, they also put together values coming from different meshes in the GP on the common mesh. I think the properties of the kernel is quite important at this point. Otherwise, the U values on the common mesh may not be comparable. This is a major assumption here that needs to be discussed.
5. There is a huge underlying smoothness assumption due to the use of GP. I believe the method assumes that small changes in the geometry will yield small changes in the simulation results. This holds only for certain problems and does not hold in others. This assumption needs to be discussed. Let me also point out that DL-based works do not even mention this. They are roughly black-box methods that do not even allow one to examine such assumptions. The proposed method is quite in good in that respect.
6. Figure 8 requires relative errors. Instead, absolute errors are given and they seem to be around 10% at certain points.

**Questions:**

Please read the weaknesses for the questions.

**Limitations:**

Limitations are not completely addressed. I noted these in the weakness section.
I do not believe there are potential negative societal impacts that should be explicitly mentioned here.

---

> ### Author Rebuttal · Authors · 2023-08-08
>
> Please refer to the general rebuttal as an introduction to our answers below.
>
> **Strengths**
>
>  3. We agree that out-of-distribution (OOD) detection is an interesting feature enabled by the predictive uncertainty. This is actually visible in some of our experiments:
>
>     - Figure 3 and 6: some values of the input pressure are OOD: they are associated with larger confidence intervals.
>
>     - Figure 19 and 20: the last 2 columns are OOD geometries: the width of the confidence intervals are much larger than for the in-distribution samples.
>
>     We will add a discussion  and carry out some experiments: we propose to use the available testing set to measure the distribution of predictive variances on non-training in-distribution samples and compare it to some OOD predictive variance.
>
> **Weaknesses**
>
>  1. We propose to complete the paragraph ending line 46 by: "Our method shares some limitations with any machine learning regressor for PDE systems: (i) within the predictive uncertainties, our method produces predictions with an accuracy lower than the reference simulations, (ii) unlike many methods used in reference simulators, like the finite element method, our method provides no guaranteed error bounds and (iii) our method requires a well sampled training dataset, which has a certain computational cost, so that the workflow becomes profitable for many-query contexts where the inference is called a large number of times. Regarding (i), rough estimates may be sufficient in preproject phases, and accuracy can be recovered by using the prediction as an initialization in the reference simulator, or by allowing the designer to run the reference simulator on the identified configuration if the regressor is used in an optimization task".
>
>  2. see 1.
>
>  3. Rigid body transformation are actually taken into account, but we recognize that it is not explicitly stated in the current version of the document:
>
>     - l.61: "Being the discretization of the support of a physical setting, the mesh inherits important features from the problem, like boundary conditions associated to some elements and points".
>
>     - l.120: "both making use of physical features inherited from the problem".
>
>     We will continue the paragraph ending line 120 by: "This means that the points, lines and  surfaces of importance in the definition of the physical problem are mapped onto their representant on the reference shape. Doing so, rigid body transformation that may occur in the database are corrected in the mesh morphing stage.".
>
>     "p" is the number of nongeometrical parameters and has been implicitly defined line 78. We will make the definition more explicit and recall it line 157.
>
>  4. In our understanding, the Jacobian of the coordinate transformation appear when a change of coordinate occur, for instance if one wants to solve the PDE directly on the common domain or compute some integral quantity with respect the fields of interest on the common domain. In the former, methods that rely on minimizing the residual of the equations on the reference domain (either for direct numerical simulation or at collocation point like in PINNs) would need to encumber themselves with this. The latter occurs when applying the PCA (especially the snapshot-POD detailed in B.2), since the computation of the correlation between two samples involves integrating over the reference domain: we agree on this part. The integral of the product of two fields of interest on an input geometry can be written on the common domain using a change of variable, and the determinant of the Jacobian of the corresponding transformation appears. In our case, the samples are supported on different geometries, so that the integral on the "input geometry" is not defined. It may be possible to use the morphing from input geo 2 to the common domain and the inverse of the morphing from input geo 1 to the common domain to write the correlation of the first two fields on input geo 1, and then write the integral on the common domain, but complex Jacobians appear, and we may lose the symmetry of the correlation operator. We will add some elements on our assumption in the paper, but in our opinion, its resolution goes beyond the scope of this paper.
>
>     We recall that in our case, we simply transport the values of the fields (coordinates and fields of interest) from the input mesh to the morphed mesh. We believe that the values coming from the different meshes can be compared to a certain extend in our setting, at least the enforced boundary conditions, and more efficiently if the morphing is not taken a priori but chosen to maximize the PCA compression (see the section **Principal component analysis** of the general rebuttal).
>
>  5. We agree that in our numerical experiences, the choice of the Matern 5/2 kernel has a consequence on the regularity of the prediction with respect to the scalar parameters and the general coordinates of the PCA decomposition of the input coordinate fields. We complete our answer by referring to the general rebuttal: **Theoretical elements** for the approximation abilities of kernels, and first paragraph of **Principal component analysis** for our futur work that may improve the regularity of the data to be approximated by the Gaussian Process. These elements will be added to the paper to address.
>
>     In our first case, the shape variations are small, but this case is representative of industrial compressor blades optimization processes. The fluid is transonic, and shocks are presents in the field of interest. The pressure field is shown on these geometries: due to the sensitivity of the outputs with respect to the inputs in this problem, the position of the pressure shock features visible variations, even under these small geometrical variations. In our opinion, this case already illustrates some level of success for MMGP for output functional with poor regularity.
>
>  6. We will update Figures 8, 21 and 22 to indicate relative errors.

---

> > ### Comment · Reviewer_4CEk · 2023-08-15
> > **thanks**
> >
> > Thanks for the detailed responses. I hope the paper will be accepted.

---

### Official Review · Reviewer_erhK · 2023-07-06

**Soundness:** 2 fair
**Presentation:** 1 poor
**Contribution:** 3 good
**Rating:** 5
**Confidence:** 4

**Summary:**

This paper presents a method that offers advantages in terms of computational efficiency by employing modal reduction compared to full models and utilizing Gaussian processing.

However, despite these strengths, the paper falls short in several aspects.
- The writing lacks clarity and coherence, with inconsistent descriptions and key information missing.
- The organization of the paper appears disjointed as if written by different individuals without synchronization.
- Furthermore, recent related works are overlooked, outdated benchmarks are used, and there is insufficient theoretical support nor enough experiments for the proposed method.

As a result, based on these limitations, I recommend rejecting this paper in its current form. **If significant improving in writing is conducted, I will re-consider**

## After rebuttal
- The authors improved their references, presentation, and empirical evaluation; Hence I increased my score to 5

**Strengths:**

- Save computational requirements by modal reduction compared to full models
- Save computational requirements by using Gaussian Processing

**Weaknesses:**

- The writing is very confusing and casual, lacking key information and consistent descriptions:
  - Authors claim they can handle huge graphs, but their maximum mesh has about 30K nodes, which is not huge.
  - They don't provide mesh size information for the solid mechanics case.
  - Inconsistent usage of "nodes" and "vertices".
  - The meaning of "finite element interpolation" (FEM shape functions?) is not defined or explained.
  - The phrase "(line 39)...high stakes" and why it prevents deployment is unclear.
  - Line 83: "Physics-based machine learning" should be corrected to "machine learning for physics".
  - Lines 87-88: Inaccurate and unnecessary information regarding the significance of each node.
  - Line 216: What are "classical engineering curves"?
  - Caption for Figure 3: Schematic plots illustrating the appearance of the four different test case geometries are needed.
  - Section 4.3: Unclear meaning of "use case".

- The paper lacks organization and coherence in both writing and technical/experimental descriptions. It seems that different sections were written independently without synchronization:
  - Multiple duplicate subsections such as "Physical setting and dataset", "Training (Implementation)", and "Results" for each dataset should be combined into one.
  - Introduction of multi-scale GNNs is scattered across different sentences (lines 94, 97-98), lacking logical structure.

- Recent related works are missing. Papers such as these utilize morphing [1,2], and there are other recent works addressing challenges in GNN performance by multi-scale [3,4,5] that should have been mentioned.

- Outdated benchmarks are used. Although the authors mention various GNN family methods, they compare their approach with an outdated GCNN. Additional comparisons to more recent methods such as MeshGraphNet and MSGNN-Grid (mentioned in their paper but not compared) should be included.

- Insufficient theoretical support is provided for the proposed method. While formal proof is not necessary, the authors fail to discuss any theoretical foundations supporting the effectiveness of their approach.

[1] Han Gao, Luning Sun, Jian-Xun Wang, "PhyGeoNet: Physics-informed geometry-adaptive convolutional neural networks for solving parameterized steady-state PDEs on irregular domain", Link: https://www.sciencedirect.com/science/article/pii/S0021999120308536

[2] Zongyi Li, Daniel Zhengyu Huang, Burigede Liu, Anima Anandkumar, "Fourier Neural Operator with Learned Deformations for PDEs on General Geometries", Link: https://arxiv.org/abs/2207.05209

[3] Lino, Mario and Fotiadis, Stathi and Bharath, Anil A and Cantwell, Chris D. “Multi-scale rotation-equivariant graph neural networks for unsteady Eulerian fluid dynamics”. Link: https://pubs.aip.org/aip/pof/article/34/8/087110/2847850

[4] Cao, Yadi, Menglei Chai, Minchen Li, and Chenfanfu Jiang. "Efficient learning of mesh-based physical simulation with bi-stride multi-scale graph neural network.". Link: https://openreview.net/forum?id=2Mbo7IEtZW

[5] Chen, Runfa and Han, Jiaqi and Sun, Fuchun and Huang, Wenbing. "Subequivariant Graph Reinforcement Learning in 3D Environments". Link: https://arxiv.org/abs/2305.18951

**Questions:**

See the writing part in **Weaknesses**

**Limitations:**

- Limited to geometries with the same topology but different transformations or deformations.
- The use of PCA, known for encoding linear correlations, poses challenges when dealing with highly complex geometries. Even the preprocessing step becomes challenging in such cases.
  - Specifically, the method is only suitable for geometries that can be compressed effectively using a cost-effective PCA approach. This contradicts the paper's claim of being applicable in real engineering use cases.

---

> ### Author Rebuttal · Authors · 2023-08-08
>
> Please refer to the general rebuttal for a discussion about dimensionality reduction, benchmarking against MeshGraphNets, limitations and theory.
>
> We will do our best to improve the writing of the paper. We would like to mention to the reviewer that with the current reviewing process, the final decision is made before the author have to opportunity to upload a new version of the paper.
>
> **Weaknesses**
>
> - Very confusing and casual writing:
>
>    - We mentioned "large" meshes, and in our third experiment, we indicate line 262 that the meshes have approximately 180k vertices.
>
>       In the second and third cases, we actually used the snapshot-POD. Line 299, we cited an article where the authors have applied the POD (also similar to PCA) to problems with millions of degree of freedom. We acknowledge that our sentence do not state that the author use POD and we will improve our writing. In [*], the author applied POD to problems up to hundreds of millions of degrees of freedom.
>
>       With such dimensionality reduction techniques, MMGP can be used with large meshes, where graph neural network method can struggle with moderate size meshes: in [10], the authors that constructed the database associated to our third case were obligated to coarsen the meshes for their GPUs to handle them.
>
>    - We will add mesh size information for the solid mechanics case: ranging between 7k to 11k.
>
>    - We will homogenize the vocabulary between "nodes" and "vertices" and carefully proof read the manuscript for other discrepancies in vocabulary and notations.
>
>    - The "finite element interpolation" is actually described in the paragraph "Finite element interpolation on a common mesh" starting line 121. We will do our best to refer the reader to this paragraph when "finite element interpolation" is mentioned elsewhere and better identify within this paragraph the very operation that corresponds to the finite interpolation.
>
>    - "(line 39)...high stakes": we will rephrase to improve our message. We meant to say that models for which the predictive uncertainty cannot be provided accurately are less likely to be used in industries where the safety standards are very high, like the aeronautical industry, and design solutions have to be provided with a high level of confidence.
>
>    - We will replace "Physics-based machine learning" with "machine learning for physics"
>
>    - We will remove the sentences: "Each node v ∈ V and each edge e ∈ E of the graph G […] that corresponds to the cartesian coordinates of each node"
>
>    - We will describe more precisely what is meant by "classical engineering curves". For instance, in Figure 3, left plot: compression rate versus massflow characteristic curve ; right plot: isentropic efficiency versus massflow characteristic curve. These kind of curves are typically used by aeronautical engineers for designing turbine blades.
>
>    - Caption for Figure 3: we will add schematic plots showing the appearance of the four used geometries. On Figure 2 of the "1 page rebuttal Figures pdf", these four geometries are shown from left to right in the following fashion: the first one shows the mesh, and the second to last show a superposition of the corresponding geometry and the mesh of the first one. The shape variations may appear small, but this case is representative of industrial compressor blades optimization processes. The pressure field is shown on these geometries: due to the sensitivity of the outputs with respect to the inputs in this problem, the position of the pressure shock features visible variations, even under these small geometrical variations.
>
>       We will also add illustrations of the four meshes used in Figure 6, with Figure 3 of the "1 page rebuttal Figures pdf"
>
>    - We will replace the four mentions of "use case"
>
>      - 4.3 section title: "AirfRANS NACA use case" -> "AirfRANS NACA dataset"
>
>      - Line 258: "a detailed description of the use case and dataset" -> "a detailed description of the physical setting and dataset"
>
>      - Line 402: In the AirfRANS NACA use case of Section 4.3 -> In Section 4.3
>
>      - D.2 section title: "AirfRANS NACA use case" -> "AirfRANS NACA dataset"
>
> - The paper lacks organization and coherence:
>
>    - We will combine the subsections "Physical setting and dataset", "Training (Implementation)", and "Results" into one
>
>    - We will rewrite the subsection "Related works" (line 83 to 100) to improve its clarity, and in particular the introduction of multi-scale GNNs, see next point.
>
>  - We will complete the "Related work" section with, following the numbered references provided by the reviewer:
>
>    - Improvements of MeshGraphNet include [3] where multi-scale and rotation-equivariant  GNN are used to extrapolate the time evolution of the fluid flow, and [4] which proposes a novel pooling strategy that prevents loss of connectivity and wrong connections in multi-level GNNs.
>
>    - Graph neural networks have been used in [5] with reinforcement learning to learn a shared policy that guides the locomotion of different agents in 3D environments
>
>    - In [1,2], the authors use deformations of irregular meshes into a reference one to learn solution of PDEs, but rely on complex coordinate transformation to compute a physical residual-based loss in the reference domain or on databases of samples of same size, and do not provide predictive uncertainties.
>
>  - Concerning outdated benchmarks: Please refer to the general rebuttal for a discussion about benchmarking against MeshGraphNets.
>
>  - Concerning the insufficient theoretical support, please refer to the general rebuttal
>
> [*] S. Grimberg, C. Farhat, R. Tezaur and C. Bou-Mosleh, Mesh sampling and weighting for the hyperreduction of nonlinear Petrov-Galerkin reduced-order models with local reduced-order bases
>
>
> **Questions**
>
> See our answers in **Weaknesses**
>
> **Limitations**
>
> Please refer to the general rebuttal for a discussion about dimensionality reduction using PCA and limitations.

---

> > ### Comment · Reviewer_erhK · 2023-08-10
> > **Reply to rebuttal**
> >
> > Thanks the authors are very keen to resolve most of my concerns, most of which are related to clear writing. I am confident they will improve in the final revision.
> >
> > Hence I would increase my score to 5. I apologize for not being able to edit the original comments (maybe because of passing due).

---

> ### Comment · Reviewer_erhK · 2023-08-10
> **Change of review results**
>
> Increased to 5. See reply to rebuttal.

---

### Official Review · Reviewer_Cr5H · 2023-07-06

**Soundness:** 3 good
**Presentation:** 3 good
**Contribution:** 2 fair
**Rating:** 6
**Confidence:** 5

**Summary:**

Current work developed a novel framework combining finite element interpolation, Gaussian process and dimension reduction techniques. It doesn't require the use of a graph network and can be trained on a CPU. The result is demonstrated on three industrial design optimization problems and two of them can beat the baseline graph neural networks.

**Strengths:**

Originality: The model is novel in case of combining the three well-known blocks of finite element analysis, Gaussian process algorithm and dimension reduction techniques without the need to use a graph neural network.

Clarity: The writing is overall clear, with minor typos.

Quality: The result demonstrated for the first two cases can outperform baseline graph neural network models.

Significance: The method is applied to several industrial design cases.

**Weaknesses:**

Quality: The third case, airfoil design is not compared with any baseline models. The error plots indicate the maximum error is large. Moreover, for the first two cases, a more advanced baseline model, like MeshGraphNet, need to be included to demonstrate the performance. More details in the question part.

Significance: The author only presents the simplest combination of the three blocks. For example, P1 finite element, Gaussian process and PCA for dimension reduction and assume it only works on the steady problem, more details in questions part.

**Questions:**

1. The reviewer has two concerns about the baseline selection. The first one is for the third case, there is no baseline to compare with, and the maximum error is large. The second concern is for the first two cases, the baseline already works very well for most of the evaluated metrics. For example, in Table 1, the baseline is already rated over 0.99 for all the metrics. In Table 2, only two baseline metrics are lower than 0.95. Though the proposed work is more computationally efficient in the demonstrated problems, the baseline graph neural network is more flexible since it can also handle unsteady problems. Therefore, it is hard to say that the developed model is better in practical applications. For the first concern, the reviewer would suggest adding more advanced baselines, like MeshGraphNet in "LEARNING MESH-BASED SIMULATION WITH GRAPH NETWORKS" for all three cases. For the second concern, the author would need to justify why the developed model is generally better than graph neural networks, either using theory or experiment.

2. The second problem is about the framework. The novel part is the view of an innovative combination of blocks of classical methods. However, inside each block, only some simple variants are chosen,  P1 finite element, vanilla Gaussian process and PCA. As the authors listed, there could be other variants for each block, but they are not tested. The review concerns that it is not a trivial extension to include other complex variants and scale the framework to more complex engineering design problems. For example, for the airfoil design, when the design condition is complex, like supersonic/hypersonic flow, the simple PCA analysis is not likely to work and requires more advanced reduction techniques, like hyper reduction listed in Ref [11]. Moreover, the P1 basis is usually not accurate enough for complex problems.\\

3. Some typos: Line 292: "the first raw" should be "the first row".

**Limitations:**

The author assumes the model work on the steady problem, fixed topology.

---

> ### Author Rebuttal · Authors · 2023-08-08
>
> Please refer to the general rebuttal for a discussion about dimensionality reduction, benchmarking against MeshGraphNets, limitations and theory.
>
> **Weaknesses**
>
> See our answers in **Questions**, where the weaknesses provided by the reviewer are addressed in more details.
>
> **Questions**
>
>  1. Third case:  we have already started to train our GCNN on this dataset and we are confident that a comparison will be available on time. The fact that we did not provide the comparison for this case is explained by the fact that the meshes are too large to be used by our GCNN implementation (requiring too much GPU memory), and need some coarsening pretreatment, as the authors in [10] were obligated to do as well. We already have derived a working coarsening pretreatment. Concerning the maximum error, if the reviewer mentions the results in the left plot of Figure 7 (for the scalar output drag coefficient C_D), we precise that although the error bars can be large, the predictions (the dots) are actually reasonably accurate. In this case, C_D is much more irregular than C_L (lift coefficient), with a learning task more difficult. We notice on the right plot of Figure 9 that the predictions of MMGP for C_L are very accurate. In [10, Figure 3], the author have compared various surrogate models for C_D and C_L, but with a strategy where these scalars are post-treated from predicted field. In our opinion, doing so is more difficult than predicting directly the scalars. We could compare from their results but in a disclaimer published on May 26, 2023, the authors indicate that they have a bug in their implementation. We propose to keep only a comparison with respect to our GCNN implementation.
>
>       Concerning the baseline comparison in the first two cases, the Q2 coefficients may be high without the results being very accurate. For instance in the third experiment, the Q2 for C_D is 0.9861 while the prediction vs accuracy plot (left plot of Figure 7) appeared reasonably accurate, but much less accurate than the predictions for C_L (right plot). Hence, all the results where the Q2 for the GCNN is lower than 0.98 (two scalars in the first case, as well as 1 scalar and 5 fields for the second case) are actually baseline where the results are not very accurate. To confirm this, we propose to add RMSE for the predictions of the first two cases.
>
>  2. We agree that, depending on the variant in each block, the scaling to complex industrial problem may become challenging design problems, and the reviewer mentions PCA and  P1 finite element interpolation.  Concerning the PCA and the morphing strategy, please see the general rebuttal.
>
>        We would like to mention that our first case is actually very close to industrial computation for designing compressor or turbine blades. The fluid is transonic, and shocks are present in the field of interest, as mentioned in line 213 and illustrated in Figure 2. An interesting feature of MMGP is the predictive uncertainty quantification, and in this case, the predictive variance is large close to the shock, meaning that MMGP is uncertain about its prediction of the shock position.
>
>       Regarding the degree of the finite element approximation, it can actually be directly inherited from the physical solver used to generate the database. For our CFD simulations (first and third cases), the data is provided at the mesh vertices, which corresponds to P1 finite elements. For the second case, the physical solver uses P1 finite elements. If a new physical setting requires higher order finite elements, it is actually straightforward in our numerical tools to compute the finite element interpolation with higher orders. On Figure 1 of the "1 page rebuttal Figures pdf", we have included an illustration for the finite element interpolation error on a sample from the testing set of the second case. We see that the finite element interpolation error is much smaller than the 90% confidence interval.
>
>  3. Thank you, the provided typo has been corrected, and the manuscript has been carefully proof read
>
> **Limitations**
>
> Please refer to the general rebuttal for a discussion about limitations.

---

> > ### Comment · Reviewer_Cr5H · 2023-08-20
> > **Reviewer response**
> >
> > Thanks for the authors' rebuttal to my questions. I still have concerns and need more clarification. Firstly, is the baseline for the third case ready now? What does the result look like? Concerning the maximum error, I am actually referring to the error contour in Figure 8. The actual UY and the error of UY are of the same order (50 vs. 30). Starting from this point, I am concerned that if the error should be that large. Secondly, the author mentioned the predictive UQ for the developed model. And where does this uncertainty come from? Does it quantify epistemic uncertainty or aleatoric uncertainty or both? What benefit can we get from the UQ. I expect it should be a good indicator of error. However, in Fig 2, the pattern of the variance, which I think is how you evaluate UQ, doesn't look like the error contour. The minor point is that if the limitation of the barycentric mapping/rbf mapping is mentioned. Can they map a donut shape geometry to a circle?

---

> > > ### Author Response · Authors · 2023-08-20
> > > **Reply to reviewer additional questions**
> > >
> > > Thank you for your comments. We have already updated the paper with the improvements we promised in the rebuttal - except for GNN/MGN additional results still pending. We have computed Q2 and relative RMSE for MMGP, with error bars, in the 3 experiments. We also have improved the finite element interpolation accuracy for meshes featuring highly distorted elements, which is the case of the third case in the boundary layer: Q2 for fields have significantly improved: from 0.9162, 0.9293 and 0.9887 to 0.9749, 0.9806 and 0.9934 for respectively Ux, Uy and p.
> > >
> > > 1. The baseline is not ready yet, although coarsening pretreatments of the meshes have been derived and applied. We also started training MeshGraphNets (MGN) for the first two experiments using the architecture from [26], Adam optimizer, MSE objective function and L2 regularization. Our results are not satisfactory yet – which confirms that training GNN and MGN are much harder for us than MMGP.
> > >
> > > 2. To us, illustrating approximation errors for fields is not well defined. Take the example of two 1D Heaviside functions slightly shifted from each other: the absolute error field would be zero everywhere except 100% in the small shift area; the relative error field would not be defined in the area where both functions are zero and be not defined or 100% in the shift area, depending on which Heaviside is the reference. When applying PCA for MMGP or optimizing (R)MSE for GNN/MGN, results are considered accurate when L^2(\Omega) approximation errors are low. Such metrics forgive large errors occurring in small areas.
> > > In short, large errors can occur locally while keeping MSE and PCA error small, and lead to predicted field featuring large local relative errors, which are undefined when reference field is zero.
> > >
> > >    In the third experiment, as seen in Figure 14, the mesh elements are very small close to the airfoil, for the physical solver to compute accurately the thin boundary layer. The boundary condition at the surface of the airfoil is zero velocity. Hence, large prediction errors can occur in the boundary layer even with accurate PCA or GNN/MGN MSE.
> > >
> > >    We updated Figures 8, 21 and 22 with relative error fields, by dividing ‘prediction-reference’ by the max of the reference field. For the two test samples 787 and 430 the worst local relative error is respectively 0.69 for Uy and 0.16 for Ux. These errors occur in such small areas that they cannot be seen on the figures, even in the local zoom provided on the trailing edges. The relative RMSE are still good, at 3x10-5 for Ux and Uy.
> > >
> > >    Finally, in both MGN papers [12] and [26], no error fields are illustrated.
> > >
> > > 3. Gaussian process regression (GPR) belongs to Bayesian methods, where the inputs are assumed to be realizations of some prior probability and the prediction corresponds to the expectation of the derived posterior probability on the outputs. In the particular case of GPR, prior and posterior are (multivariate) normal distributions, and the predictions involve a chosen kernel, which hyperparameters are then calibrated during training. In our case, the setting is deterministic (coming from resolution of deterministic physics solvers) so that no aleatoric uncertainties exist, only epistemic uncertainty. The uncertainty comes from the fact that from new input values (mesh and nongeometry uncertainty in our case), under GPR assumptions, the output is provided under the form of a normal distribution, defined by its expectation and variance. We can then quantify the probability of the output to lie in a so-called confidence interval, under GPR assumptions. In real life configurations, the inputs do not follow the assumed prior probability, so the provided predictive uncertainties are not guaranteed (although an empirical verification is provided in Table 3, and they appear quite accurate).
> > >
> > >     The benefit that we can get from predictive uncertainties are:
> > >
> > >       - identify out-of-distribution geometrical configurations,
> > >
> > >       - provide estimates for confidence intervals which, although not guaranteed, can help engineers identify more trustworthy models.
> > >
> > >     We agree that in Figure 2, the variance and error contours are not identical, althoufh largest values of each lie on the pressure shock. Error and variance are linked, but not equivalent. The variance indicates the magnitude of the possible predicted values: the reference value may be close to the expectation of MMGP output distribution, the predictive error can be zero even if the confidence interval is large.
> > >
> > > 4. In our implementation, donut shape geometries cannot be mapped into a disk, since the input and target shape do not have the same topology. Mesh deformation algorithms compatible to topology changes do exist [*], but since we did not explicitly use one of them in our experiments, we decided to assume fixed topology.
> > >
> > > [*] A. Zaharescu et al. Topology-Adaptive Mesh Deformation for Surface Evolution, Morphing, and Multi-View Reconstruction, 2010

---

> > > > ### Comment · Reviewer_Cr5H · 2023-08-20
> > > >
> > > > Thanks for the elaboration on UQ and mesh morphing. I am satisfied with the explanations. However, to have a learning based baseline for the third case is a key component for a complete work. I encourage the authors to include one and compared with MMGP before the deadline. I will adjust my evaluation once the new baseline is provided.

---

> > > > > ### Author Response · Authors · 2023-08-21
> > > > > **Baseline comparison for third case**
> > > > >
> > > > > Dear reviewer,
> > > > >
> > > > > In the remaining time before the deadline (today 1 pm EDT), we were able to compare accuracy estimators for lift and drag coefficients CL and CD with the four estimators produced by the authors of the airfRANS dataset [a], in the setting “full dataset”, with their proposed loss function [a, eq (3)], and with their recently updated results.
> > > > >
> > > > > Please keep in mind that we did not use the same meshes as in [a], since they required to coarsen the meshes for the training to be doable on GPUs, whereas we kept the full meshes with MMGP. Contrary to the direct prediction of the CL and CD coefficient in the original version of our paper, we recomputed them in the following results by integrating the reference and MMGP predicted wall shear stress (from the velocity) and pressure fields around the surface of the airfoil, in the same fashion as [a] did for their predictions. We did not use the exact code routine from the github of [a], but our recomputed scalars are coherent with the ones from [a].
> > > > >
> > > > > The models from [a] are a MLP (a classical Multi-Layer Perceptron), a GraphSAGE [b], PointNet [c] and Graph U-Net [d]. The use of a MLP is possible because, to our understanding, the data is taken at a fixed number of 32,000 nodes uniformly sampled in the computational domain. Please refer to [a, appendix L] for a description of the architecture used. For MMGP, we use our 10 trained models, as described in Section 4.3. The results are provided in the Table below (the baselines being taken from [a, Table 19]:
> > > > >
> > > > > *Relative errors*
> > > > >
> > > > > **CD**
> > > > >
> > > > > MLP: 6.18 +/- 0.90
> > > > >
> > > > > GraphSAGE: 7.37 +/- 1.21
> > > > >
> > > > > PointNet: 17.4 +/- 1.4
> > > > >
> > > > > Graph U-Net: 13.3 +/- 0.9
> > > > >
> > > > > MMGP : **0.760 +/- 3.50e-4**
> > > > >
> > > > > **CL**
> > > > >
> > > > > MLP: 0.21 +/- 0.03
> > > > >
> > > > > GraphSAGE: 0.15 +/- 0.03
> > > > >
> > > > > PointNet: 0.20 +/- 0.0.03
> > > > >
> > > > > Graph U-Net: 0.17 +/- 0.02
> > > > >
> > > > > MMGP : **0.0289 +/- 3.96e-5**
> > > > >
> > > > > *Spearman’s correlation*
> > > > >
> > > > > **CD**
> > > > >
> > > > > MLP: 0.25 +/- 0.09
> > > > >
> > > > > GraphSAGE: 0.19 +/- 0.07
> > > > >
> > > > > PointNet: 0.07 +/- 0.06
> > > > >
> > > > > Graph U-Net: 0.09 +/- 0.05
> > > > >
> > > > > MMGP : **0.718 +/- 1.38e-4**
> > > > >
> > > > > **CL**
> > > > >
> > > > > MLP: 0.9932 +/- 0.0017
> > > > >
> > > > > GraphSAGE: 0.9964 +/- 0.0007
> > > > >
> > > > > PointNet: 0.9919 +/- 0.0017
> > > > >
> > > > > Graph U-Net: 0.9949 +/- 0.0011
> > > > >
> > > > > MMGP : **0.9992 +/- 2.205e-6**
> > > > >
> > > > > We still plan to update our results for the camera-ready version of the paper with comparisons with MeshGraphNets for the three cases and our GCNN implementation for the third one. Finally, our goal is not to outperform every existing GNN, but rather to propose an efficient and easy-to-train alternative on our chosen perimeter of assumptions, with interesting properties.
> > > > >
> > > > >
> > > > >
> > > > > [a] F. Bonnet, J. Mazari, P. Cinnella, and P. Gallinari. Airfrans: High fidelity computational fluid 596 dynamics dataset for approximating Reynolds-Averaged Navier–Stokes solutions. In S. Koyejo, 597 S. Mohamed, A. Agarwal, D. Belgrave, K. Cho, and A. Oh, editors, Advances in Neural Information Processing Systems, volume 35, pages 23463–23478. Curran Associates, Inc., 599 2022.
> > > > >
> > > > > [b] W. Hamilton, Z. Ying, and J. Leskovec. Inductive representation learning on large
> > > > > graphs. In I. Guyon, U. Von Luxburg, S. Bengio, H. Wallach, R. Fergus, S. Vishwanathan,
> > > > > and R. Garnett, editors, Advances in Neural Information Processing Systems, volume 30. Curran
> > > > > Associates, Inc., 2017.
> > > > >
> > > > > [c] R.Q. Charles, H. Su, M. Kaichun, and L.J. Guibas. Pointnet: Deep learning on point sets
> > > > > for 3d classification and segmentation. In 2017 IEEE Conference on Computer Vision and Pattern
> > > > > Recognition (CVPR), pages 77–85, 2017. doi: 10.1109/CVPR.2017.16.
> > > > >
> > > > > [d] H. Gao and S. Ji. Graph U-Nets. In Kamalika Chaudhuri and Ruslan Salakhutdinov,
> > > > > editors, Proceedings of the 36th International Conference on Machine Learning, volume 97 of Proceedings of Machine Learning Research, pages 2083–2092. PMLR, 09–15 Jun 2019.

---

> > > > > > ### Comment · Reviewer_Cr5H · 2023-08-21
> > > > > >
> > > > > > Thanks for providing the additional baseline on time. It is more convincing now that the MMGP is a good model. Because of the explanations provided on my questions as well as the additional baselines, I raised my scores to 6 and would support acceptance. Please be sure to include more metrics in your future revision for the third dataset, as you said. One minor point is that I believe Spearman’s correlation should be $\rho_{D}$ and $\\rho_{L}$, please revise it.

---

### Official Review · Reviewer_iwUN · 2023-07-09

**Soundness:** 3 good
**Presentation:** 3 good
**Contribution:** 3 good
**Rating:** 6
**Confidence:** 2

**Summary:**

The paper proposes a framework for solving physical problems defined over discretized geometrical spaces, such as meshes. While prior methods like graph neural networks have been the go-to tool for solving such problems, they need large datasets to train and don't output predictive uncertainties out of the box. This paper, thus takes a mesh morphing approach (assuming fixed topology) to project various meshes to a common support followed by classical dimensionality reduction techniques and Gaussian processes. This allows for dealing with large meshes, without knowing any parameterization and yields predictive uncertainties.

**Strengths:**

The major strength of this work lies in the fact that it uses classical techniques such as Gaussian processes, PCA and mesh morphing to solve a key problem in learning-based simulation research which is the need for large datasets and lack of generalization. The paper cleverly uses the mesh morphing idea to eliminate the need to learn shape characteristics and focus purely on the features required to solve the physical problem. It is also interesting to note the use of the finite element method to arrive at common support, which again provides a fair degree of inductive bias to this framework leading to a higher generalization without needing a large dataset to train with, as is evident via a significant agreement between GCNN and MMGP in Figure 1. The results presented in the paper are sufficient to demonstrate the utility of this method.

**Weaknesses:**

Perhaps the biggest limitation of this method arises from the fact that it assumes a fixed topology. Since most domains are not so simple and homogeneous, I wonder if this is going to cause issues when this work is applied more generally.

**Questions:**

NA

---

> ### Author Rebuttal · Authors · 2023-08-08
>
> Please refer to the general rebuttal as an introduction to our answers below.
>
> **Weaknesses**
>
> From our experience and information, the fixed topology limitation still falls into a very broad range of industrial applications, for instance optimizing compressor and turbine blades.
>
> We still mention that mesh deformation algorithms compatible to topology changes do exist [*], but since we did not explicitly use one of them in our experiments, we decided to assume fixed topology. We will add this reference to the paper.
>
> We precise that the morphing used in this work (Tutte's barycentric mapping and radial basis function morphings - RBF) can deal with complex and non-homogeneous domains.
>
> For instance in [23], which is a recent evolution of a family the algorithms that emerged from Tutte's barycentric mapping, extremely complex geometries and even very poor quality meshes are handled with high quality results. [**] presents results of RBF morphing of complex 3D surfaces.
>
> We produced a method that is already competitive in our experiments, using basic morphing algorithm, but we agree that the use of more advanced morphings is definitively an axis of research: we will complete the end of the conclusion to indicate these perspectives. We are already working on morphing algorithms optimized to minimize the number of PCA modes.
>
> [*] Andrei Zaharescu, Edmond Boyer, and Radu Horaud, Topology-Adaptive Mesh Deformation for Surface Evolution, Morphing, and Multi-View Reconstruction: https://arxiv.org/abs/2012.05536
>
> [**] Mario Botsch and Leif Kobbelt, Real-Time Shape Editing using Radial Basis Functions
>
> **Questions**
>
> N/A
>
> **Limitations**
>
> N/A

---

> > ### Comment · Reviewer_iwUN · 2023-08-13
> > **response to rebuttal**
> >
> > Thanks for the rebuttal. I would like to keep my original score.

---

### Official Review · Reviewer_mqry · 2023-07-26

**Soundness:** 3 good
**Presentation:** 3 good
**Contribution:** 3 good
**Rating:** 5
**Confidence:** 2

**Summary:**

This paper presents a method for solving physics problems in design optimization, with scalar and non-parameterized mesh as inputs. One main contribution of the work is to use a mesh morphing pretreatment with finite element interpolation. The other idea is to leverage shape embedding by dimensional reduction of coordinates that are seen as continuous fields over the geometrical support.
This work seems to be able to handle large meshes and is easily trained on CPU with good interpretability.

**Strengths:**

- The idea of using mesh morphing pretreatment with finite element interpolation seems to be effective as shown in the experiments.
- The technique of using the shape embedding by dimensional reduction reduces the computational cost and enables efficient computation on CPU.
- The results look nice.

**Weaknesses:**

I'm not an expert in this field and not familiar with the latest advances in physical simulation.
One weakness I can think of maybe that dimensionality reduction using PCA seems to be pretty standard and commonly used in all kinds of applications. If this is the main contribution, the technical contribution may be limited.

**Questions:**

I'm not an expert in this field. Just curious, is it possible to use the particle-based method like MPM to solve the physics problems shown in this paper? If yes, what is the advantages of the proposed method?

**Limitations:**

No limitations are provided.

---

> ### Author Rebuttal · Authors · 2023-08-08
>
> Please refer to the general rebuttal for a discussion about dimensionality reduction, benchmarking against MeshGraphNets, limitations and theory.
>
> **Weaknesses**
>
> The two main contributions of our work are:
> - A deterministic and data pretreatment for formatting variable-size samples on a common fixed-size description: these are the mesh morphing and finite element interpolation steps.
> - A deterministic shape embedding by dimensionality reduction of the coordinates of the mesh vertices interpreted as continuous fields of the common geometrical support.
>
> Advantages of relying on deterministic operations, handling reduced size objects and gaining predictive uncertainty quantification are provided in the general rebuttal.
>
> We also refer to the **Principal component analysis** section of the rebuttal to detail how our use of PCA leads to nonlinear dimensionality reduction and how it can be improved.
>
> **Questions**
>
> The method presented in this work aims to approximate solutions of physics problem by computing a surrogate model, using a classical training phase. During the training phase, a dataset of simulated physical solution is constructed: any numerical simulation method can be used for this purpose. In Section 4.1 (three-dimensional fluid dynamics case) we use the elsA solver [*] with the finite volumes method for solving the compressible steady-state Reynolds-Averaged Navier-Stokes equations, in Section 4.2 (two-dimensional solid mechanics) we use the Z-set solver [**] with the finite elements method for solving the continuum mechanics equations in a quasistatic setting with non-linear materials, and in Section 4.3 (AirfRANS NACA use case), the authors of [10] use simpleFOAM for solving the two-dimensional incompressible steady-state Reynolds-Averaged Navier-Stokes equations.
>
> The reviewer may think of [A], where GNN are used to construct surrogates problems for particle-based simulations. In this case, to construct a graph on the simulated data, some notion of neighboring points is considered. Our method heavily relies on meshes, but is it still possible to preprocess particle-based simulations to project solutions onto meshes, in a similar fashion as [***] constructed graphs of the data to train GNNs.
>
> To summarize our answer: particle-based methods can be used to generate the dataset, and our method produces a surrogate model trained on this dataset.
>
> [*] Cambier, Laurent and al. "An Overview of the Multi-Purpose elsA Flow Solver." Aerospace Lab (2011)
>
> [**] http://www.zset-software.com
>
> [***] Sanchez-Gonzalez, Alvaro et al. “Learning to Simulate Complex Physics with Graph Networks.” ArXiv abs/2002.09405 (2020)
>
> **Limitations**
>
> Please refer to the general rebuttal for a discussion about limitations.

---

> > ### Comment · Reviewer_mqry · 2023-08-17
> > **Post rebuttal**
> >
> > Thanks for the rebuttal! The rebuttal has addressed my concerns. I will keep my previous score.

---

### Author Rebuttal · Authors · 2023-08-08

We first remind of these general points:

- MMGP handles nonparametrized meshes (of variable size and connectivity) and scalar parametric variability within assumptions verified by a large set of industrial applications
- MMGP does not rely on deep learning, but on well known Gaussian Processes on reduced dimension input and ouput settings. Our goal is not to outperform every existing GNN, but rather to propose an efficient and easy-to-train alternative on our chosen perimeter of assumptions, with interesting additional properties
- MMGP provides predictive uncertainties at limited cost, enabling efficient identification of out-of-distribution parameters and geometries
- MMGP makes use of important features inherited from the meshes: points, lines and surfaces of importance in the definition of the physical problem are mapped onto their representant on the reference shape. Doing so, rigid body motions are corrected in the mesh morphing stage, and fields of interest values at boundary conditions are mapped to the same location on the geometry of reference

Then, we address four key points pointed out by several reviewers

**Principal component analysis**

In our work, PCA is applied after morphing and finite element interpolation of the data, which results in a nonlinear dimensionality reduction (dim red). For now, the morphing technique is chosen a priori. Future work consists in optimizing the morphing to minimize the number of PCA modes, so that the overall nonlinear dim red stage can become highly nonlinear and adapted to the data. Our intuition: even in pathological cases of transported discontinuous fields, the morphing may align the discontinuities at the same positions on the geometry of reference, so that the PCA can be accurate with few modes, and the predicted quantities of interest can recover some regularity with respect to the general coordinates of the PCA decomposition.

In experiments 2 and 3, we used the snapshot-POD dim red (see supplementary material l.420). This method takes into account inhomogeneity of the mesh by giving a weight to the data in the mesh proportional to the size of the corresponding element. The snapshot-POD can be easily parallelized on various computer nodes with distributed memory tasks, and we have carried it out on meshes up to 2 million nodes in other projects.

**Comparisons with MeshGraphNet (MGN)**

Unet-type GNN is a classical baseline that can be outperformed by more recent architectures such as MGN, which are designed for learning unsteady simulations: the solution at the next time step is predicted from current time step. In this setting, input and output are very close to each other, as enforced by the time stepping scheme of the physical solver used to generate the database. In our work, we consider steady problems. MGNs can be applied for steady problems by considering a single large time-step to predict the stationary solution, like in the original paper [*, appendix A.4.2] and in [**]. To our understanding, the steady problems considered in these papers do not contain geometrical variabilities: a single NACA airfoil is used to generate the data and the samples only differ in terms of scalar-valued parameters. This setting is simpler than what we considered in our 3 experiments.

We see no obvious difficulty for implementing MMGP to learn time-dependent simulations, although we did not try it.

[*] T. Pfaff et al (2020). Learning mesh-based simulation with graph networks

[**] L. Harsch et al (2021). Direct prediction of steady-state flow fields in meshed domain with graph networks

**Limitations**

Limitations (provided in Section 2) are
- The geometry is controlled and assumed without noise
- The considered geometrical transformations do not contain extreme distortions
- The topology is fixed
- The scalars and field of interest are stationary

From our experience, these limitations, in particular the last two, still encompass a broad range of industrial applications. Roughly 90% of fluid flow simulations computed by aircraft engine manufacturers use stationary models (optimizing compressor and turbine blades fall into our assumptions)

**Theoretical elements**

We will add a section regrouping the following elements:
- Efficiency is gained from having significant part of the method treated with deterministic and interpretable tasks:
  - Data pretreatment for formatting variable-size samples on a common fixed-size description: mesh morphing and finite element interpolation
  - Shape embedding by dim red of the coordinates of the mesh vertices interpreted as continuous fields of the common geometrical support, using PCA

  Relying on deterministic operations means that the machine learning has fewer tasks to learn, and reduces variability so that fewer examples are enough to learn powerful models
- The learning is done in small input and output dimensions, instead of handling large objects, which can be hard with GNNs
- The straightforward estimate of predictive uncertainties helps gaining confidence in the prediction and identifying out-of-distribution geometries or parameters.
- Under the chosen assumptions, the solutions of interest computed by the physical solver have high quality (optimal error estimates with respect to the solution are generally available). The continuous representation of the fields of interest over the geometrical support provided by finite element interpolation is then accurate. The coordinates fields are also represented as continuous fields in the same fashion
- From [***], there exists conditions on the features of a continuous kernel so that it may approximate an arbitrary continuous target function: universal approximating property. Gaussian Processes predictions being expressed as kernel evaluations, they have the theoretical foundation to be able to approximate arbitrary continuous quantities of interest

[***] C.A. Micchelli et al (2006). Universal Kernels. Journal of Machine Learning Research, 7(12)

---

### Comment · Area_Chair_sDdo · 2023-08-13

Dear reviewers and authors,

Thank you very much for your work on this submission and its evaluation. Now that the authors have responded to the reviews, I strongly encourage the reviewers to acknowledge the review, to look at other reviews and rebuttals for this submission, and to adjust their scores if needed. Thanks to those that have already done so.

Authors have the possibility to reply if further questions are needed, until the 16th.

Thank you very much to all,
Area Chair

---

### Decision · Program_Chairs · 2023-09-21

**Decision:**

Accept (poster)

**Comment:**

The author consider an approach for constructing a GP-based surrogate model to predict results of a physical simulation depending on a geometry and/or physical parameters. The novelty is that the geometry is defined by a mesh and the model input can also include physical parameters defining the initial physical simulation. The authors proposed a procedure  to pre-process meshes (mesh morphing approach) to unify different meshes with different structures and to reduce their dimensionalities.

The paper has some weaknesses (see detailed comments of the reviewers below). The main weakness is how the text is organised. The description of the algorithm should be provided in more explicit and easy-to-follow terms. Each step is better to explain based on some simple (toy) example. It could be good to provide some graphic illustration with schematic overview of the algorithm.
The proposed method consists of several moving parts. Thus I also expect some more detailed ablation study.

At the same time, the method is very useful. There are many applied problems that can be solved using the proposed approach. The experimental results demonstrate efficiency of the proposed approach. Thus I vote to accept this paper.